# Risk of lead exposure, subcortical brain structure, and cognition in a large cohort of 9- to 10-year-old children

**Andrew T. Marshall**[1], **Rob McConnell**[2], **Bruce P. Lanphear**[3], **Wesley K. Thompson**[4], **Megan M. Herting**[2], **Elizabeth R. Sowell**[1]*

**1** Children's Hospital Los Angeles, and the Department of Pediatrics, University of Southern California, Los Angeles, California, United States of America, **2** Department of Population and Public Health Sciences, University of Southern California, Los Angeles, California, United States of America, **3** Faculty of Health Sciences, Simon Fraser University, Vancouver, British Columbia, Canada, **4** Department of Biostatistics, Department of Family Medicine and Public Health, University of California, San Diego, San Diego, California, United States of America

* esowell@chla.usc.edu

**Data Availability Statement:** The data for this study are publicly available from NIMH's National

## Abstract

### Background

Lead, a toxic metal, affects cognitive development at the lowest measurable concentrations found in children, but little is known about its direct impact on brain development. Recently, we reported widespread decreases in cortical surface area and volume with increased risks of lead exposure, primarily in children of low-income families.

### Methods and findings

We examined associations of neighborhood-level risk of lead exposure with cognitive test performance and subcortical brain volumes. We also examined whether subcortical structure mediated associations between lead risk and cognitive performance. Our analyses employed a **c**ross-sectional analysis of baseline data from the observational Adolescent Brain Cognitive Development (ABCD) Study. The multi-center ABCD Study used school-based enrollment to recruit a demographically diverse cohort of almost 11,900 9- and 10-year-old children from an initial 22 study sites. The analyzed sample included data from 8,524 typically developing child participants and their parents or caregivers. The primary outcomes and measures were **s**ubcortical brain structure, cognitive performance using the National Institutes of Health Toolbox, and geocoded risk of lead exposure.

Children who lived in neighborhoods with greater risks of environmental lead exposure exhibited smaller volumes of the mid-anterior (partial correlation coefficient [$r_p$] = -0.040), central ($r_p$ = -0.038), and mid-posterior corpus callosum ($r_p$ = -0.035). Smaller volumes of these three callosal regions were associated with poorer performance on cognitive tests measuring language and processing speed. The association of lead exposure risk with cognitive performance was partially mediated through callosal volume, particularly the mid-posterior corpus callosum. In contrast, neighborhood-level indicators of disadvantage were not associated with smaller volumes of these brain structures.

Data Archive. The ABCD data used in this report came from https://doi.org/10.15154/1523347.

**Funding:** ABCD Acknowledgement. Data used in the preparation of this article were obtained from the Adolescent Brain Cognitive Development (ABCD) Study (https://abcdstudy.org/), held in the NIMH Data Archive (NDA). This is a multisite, longitudinal study designed to recruit more than 10,000 children age 9-10 and follow them over 10 years into early adulthood. The ABCD Study is supported by the National Institutes of Health and National Institute on Drug Abuse and additional federal partners under award numbers U01DA041022, U01DA041028, U01DA041048, U01DA041089, U01DA041106, U01DA041117, U01DA041120, U01DA041134, U01DA041148, U01DA041156, U01DA041174, U24DA041123, and U24DA041147. A full list of supporters is available at https://abcdstudy.org/federal-partners/. A listing of participating sites and a complete listing of the study investigators can be found at https://abcdstudy.org/principal-investigators/. ABCD consortium investigators designed and implemented the study and/or provided data but did not necessarily participate in analysis or writing of this report. This manuscript reflects the views of the authors and may not reflect the opinions or views of the NIH or other ABCD consortium investigators. The ABCD data repository grows and changes over time. The ABCD data used in this report came from https://doi.org/10.15154/1503209.

**Competing interests:** The authors have declared that no competing interests exist.

## Conclusions

Environmental factors related to the risk of lead exposure may be associated with certain aspects of cognitive functioning via diminished subcortical brain structure, including the anterior splenium (i.e., mid-posterior corpus callosum).

## Introduction

Elevated blood-lead concentrations during childhood, which are more common among low-income children [1], are associated with intellectual deficits and behavioral problems [2–13]. While it has been estimated that 40% of the world's children have elevated blood-lead levels, 90% of these children live in low- and middle-income countries [14]. Lead-based paint and leaded gasoline reflect primary transnational sources of exposure, but there may be many more sources of lead exposure in developing countries, such as lead-glazed ceramics, lead mining and smelting, flour mills, lead-battery recycling plants, and lead-containing medicines and cosmetics [15,16]. The annual cost of childhood lead exposure in these low- and middle-income countries is nearly $1 trillion [17].

Lead undoubtedly has effects on brain development [18], but most studies investigating its effects on brain structure and function involve heavily exposed children and adults [19–24]. We recently reported that greater *risk* of lead exposure [25,26]–based on age of housing and poverty levels in participants' residential neighborhoods [1,27]–was negatively associated with cognition and whole-brain cortical structure in 9- and 10-year-old children of low-income families [28]. Cortical vertex maps showed that reductions in total cortical surface area and volume were associated with increasing lead risk in children of low- but not high-income families.

Lead exposure is also associated with subcortical brain morphology. Occupational lead exposure is inversely associated with posterior corpus callosum and periventricular white matter volumes [19], and childhood lead exposure is associated with altered white-matter connectivity in young adults' corpus callosum and corona radiata [21]. Given different spatiotemporal trajectories of white- and gray-matter development [29,30], the low-socioeconomic-status (SES)-specific inverse associations that we observed between lead risk and cortical gray matter [28] may be differentially manifested within subcortical brain regions. If the cognitive effects of lead exposure are mediated by effects on brain structures that mature at different rates and times during development (e.g., in-utero, infancy, childhood, adolescence) [18], then identifying such regional biomarkers and their associations with neighborhood-level risk of exposure may increase our understanding of the temporally-dependent impact on developmental outcomes.

No studies have investigated associations between children's low-level lead exposure and subcortical brain structure. We employed neighborhood-level risk scores, which are validated proxies for census-tract-level prevalence of elevated blood-lead levels and mean blood-lead levels [28,31], to evaluate associations between brain structure and lead risk at high and low levels of exposure. We analyzed Adolescent Brain Cognitive Development[SM] Study (ABCD Study[®]) data to determine relationships between lead-exposure risk, subcortical volumes, and performance on NIH Toolbox[®] cognitive tests [32,33]. Our goal was to explore (1) associations between subcortical brain regions and lead-exposure risk, (2) whether family income moderated these associations, (3) subcortical volumes' associations with cognitive performance, and (4) whether the volumes of those subcortical regions mediated relationships between lead risk and cognition.

## Methods

### Participants

ABCD is a 10-year longitudinal study involving 21 U.S. study sites [34]. Using school-based enrollment [35], the consortium enrolled nearly 11,900 9- and 10-year-old children from an initial 22 sites. The ABCD Study's data collection sites all used the same protocol for recruitment, testing, and neuroimaging [36]. ABCD demographics correspond well with the American Community Survey (ACS) [37]. Our data came from the July 2019 ABCD 2.0.1 data release [38], which included baseline data for 11,875 children. Currently, there are no blood-lead data for ABCD participants.

Centralized IRB approval was obtained from the University of California, San Diego. Study sites obtained approval from their local IRBs. Parents provided written informed consent; children provided written assent. Data collection and analysis complied with all ethical regulations.

### Lead risk and area deprivation index (ADI)

We used a high-resolution nationwide map (https://www.vox.com/a/lead-exposure-risk-map) to obtain geocoded lead-risk scores for participants' census tracts [25,26]. These scores reflect national deciles of a weighted sum of two validated correlates of lead exposure (https://github.com/voxmedia/data-projects/blob/master/vox-lead-exposure-risk/calculate-lead-risk.py): the ages of homes (weight = 0.58) and poverty (0.42) rates [1,27], both derived from ACS data. Lead-risk scores were previously shown to be highly associated with childhood lead exposure in children [28,31]. For instance, we previously showed that these lead-risk scores were positively associated with both the rate of elevated blood-lead levels across 13 states and 2 cities at the census-tract level and the geometric mean of census-tract-level blood-lead levels in the state of Maryland [28]. ABCD site-by-site lead-risk-score distributions have been published previously [28]. Briefly, for some sites, the lead-risk scores were uniformly distributed, while other sites showed modalities at either lower, intermediate, and/or higher lead-risk scores, thereby reflecting geographic differences in risk of exposure [39].

ADI is a composite metric of neighborhood deprivation (e.g., low education, poor plumbing) [40,41] and, like lead-exposure risk, incorporates poverty rates. Unlike lead-exposure risk, ADI does not include data related to age of housing. Census-tract-level ADI was computed based on coefficient values of past research [40] and discretized into national percentiles. The R code for computing and merging ADI (and its national percentile) with ABCD data is available: https://github.com/ABCD-STUDY/geocoding/blob/master/Gen_data_proc.R.

### ABCD data

We analyzed uncorrected baseline performance on seven NIH Toolbox tests [32] and baseline structural brain measures (volumes of 21 subcortical regions) [42]. Data collection procedures are described in detail elsewhere [32,42,43].

NIH Toolbox tests show good convergent validity compared with gold standards of cognitive testing [44]: (1) the Picture Vocabulary Test (a measure of language; Version 2.0, Ages 3+), (2) the Flanker Inhibitory Control and Attention Test (attention, executive function; Version 2.0, Ages 8–11), (3) the List Sorting Working Memory Test (working memory; Version 2.0, Ages 7+), (4) the Dimensional Change Card Sort Test (executive function; Version 2.0, Ages 8–11), (5) the Pattern Comparison Processing Speed Test (processing speed; Version 2.0, Ages 7+), (6) the Picture Sequence Memory Test (episodic memory; Version 2.0, Form A, Ages 8+), and (7) the Oral Reading Recognition Test (language; Version 2.0, Ages 3+). Briefly,

the NIH Toolbox was administered on an iPad (~25–30 min to complete), with all task's instructions being read by the examiner, except for the Pattern Comparison Processing Speed Test (i.e., presented by an audio recording). All tasks were administered with the iPad in an upright position (~45° degree angle), with some tasks incorporating a home-base "button" in front of the iPad for the participants on which the participants would place their index finger between the task's trials (Flanker Inhibitory Control and Attention Test, Dimensional Change Card Sort Test).

We obtained measures of subcortical brain structure using FreeSurfer v5.3.0 on acquired $T_1$w MRI volumes; ABCD neuroimaging data collection and processing procedures have been described [45]. Depending on the study site, Siemens, Philips, and GE scanners were used, with $T_1$ acquisition times (min:s) being 7:12, 5:38, and 6:09, respectively. Accordingly, a random effect of scanner serial number was included in analyses (as described below). All neuroimaging parameters are available: https://abcdstudy.org/images/Protocol_Imaging_Sequences. pdf. Structural magnetic resonance imaging (MRI) $T_1$ images were acquired first, followed by functional MRI and diffusion MRI images (data not shown). $T_1$w images were corrected for gradient nonlinearity distortions per scanner-manufacturer guidelines, with further details comprehensively described previously [45]. ABCD data are publicly available on the NIMH Data Archive (https://data-archive.nimh.nih.gov/abcd).

## Statistical analyses

Analyses included 8,524 children with complete data for the variables of interest (Table 1 in S1 Appendix). Participants' data were excluded if the primary residential address was invalid (remaining $n = 11,175$) or unable to be geocoded into a 1–10 lead-risk score (remaining $n = 11,169$), a valid household/family income was not provided (remaining $n = 10,234$), there were missing data for sex, age, parental education, race, ethnicity, NIH toolbox test or composite scores, or structural imaging measures (remaining $n = 9,519$), if the ADI score was missing or invalid (weighted sum = 0) (remaining $n = 9,331$), or if the neuroimaging data did not pass all quality-control measures or there were neuroanatomical variants (incidental findings) judged to be of possible clinical significance (final $n = 8,524$) [46].

We employed linear mixed-effects models to determine lead-risk associations and family income × lead-risk interactions on 21 subcortical volumes. We averaged bilateral data across both hemispheres (Table 1 in S1 Appendix). In accordance with previous research analyzing associations between brain structure and socioeconomic and/or environmental conditions [28,47,48], we controlled for children's age, sex, race, ethnicity, maximum parental education, and family income. Analyses of subcortical data also controlled for intracranial volume (ICV), which was inversely associated with both lead risk, Spearman's rho ($\rho$) = -0.11, $p < .001$, and ADI, $\rho$ = -0.11, $p < .001$. Random-effects structures included random intercepts for MRI serial number (i.e., some study sites have multiple machines) and family identification number (i.e., many ABCD participants were siblings). We used the Benjamini-Hochberg false-discovery-rate (FDR) algorithm to correct for multiple comparisons [49].

Lead risk, age (in months), and ICV were continuous factors. Maximum parental education was a continuous factor with seven levels (1 = ≤6th grade; 2 = 7th-9th grade; 3 = 10th-12th grade, no diploma; 4 = high-school graduate, GED or equivalent; 5 = Some college with no degree, Associate's degree; 6 = bachelor's degree; 7 = master's degree, professional degree, or doctorate). Children's race and ethnicity were categorical factors derived from parent reports on the child. Race had 6 levels: "White", "Black", "Asian", "American Indian or Alaska Native", "Native Hawaiian or Other Pacific Islander", or "Other" (e.g., multiracial). Ethnicity had two levels: "Hispanic" or "Not Hispanic". As within ABCD's Data Exploration and Analysis Portal

(deap.nimhda.org), family income was a categorical factor with 3 levels, per parents' reported household income (Low Income: ≤$50K; Middle Income: $50K-$100K; High Income: ≥$100K). Categorical factors were effects-coded to facilitate interpretation of main effects given higher-level interactions [50]. Continuous factors were centered (i.e., a constant was subtracted) to make parameter estimates more interpretable [50].

To evaluate whether lead-risk associations could be explained by general neighborhood disadvantage, we conducted sensitivity analyses in which ADI replaced lead risk in the models. The ADI national percentile was converted into deciles and centered to match the lead-risk analyses. To further verify lead-risk associations with subcortical volume, we replaced lead risk with age of housing (i.e., the factor included in computing lead risk but not ADI). These data, which generally reflect census-tract-level housing-age-based estimates of the proportion of homes with lead-based paint hazards [27], were maintained on their original scale (range = 0.05–64.54), Box-Cox transformed to correct for positive skewness, and mean-centered for analysis.

Because our research has shown that cortical structure mediates associations between environmental factors (family income) and cognition [47], we examined whether subcortical structure mediated associations between lead risk and cognitive performance. Subcortical volume was mean-centered when estimating indirect associations. Analyses employed linear mixed-effects models, with children's sex, age, parental education, race, ethnicity, and family income as covariates. Here, the criterion was cognitive test score, so the random-effects structure included random intercepts for study site (rather than MRI serial number) and family identification number. Accordingly, because these analyses ultimately tested whether the association between lead risk and cognitive test score was mediated by subcortical volume, in which the criterion was cognitive test score and not subcortical volume, ICV was not included as a covariate in these analyses of cognitive performance. The presence of statistically significant indirect associations (i.e., products of unstandardized regression coefficients: lead-risk→brain-structure, brain-structure→cognition, controlling for lead risk) was evaluated via construction of bias-corrected percentile bootstrapping 95% confidence intervals (CI; 10,000 bootstrapped samples) [51], irrespective of the statistical significance of the corresponding total effect [52]. To minimize issues of multicollinearity of multiple mediators (i.e., collinear relationships between subcortical volumes of different regions) [53], only subcortical volumes associated with cognitive test performance were tested as mediators.

We conducted analyses using MATLAB's Statistics and Machine Learning Toolbox 11.7 (R2020a; MathWorks). Model output and model-fit characteristics are provided in Tables 2–116 in the S1 Appendix. Statistical reporting in the main text is in the form of $t$-tests except when involving categorical factors (e.g., family income), in which results are in the form of $F$ tests computed using MATLAB's *anova* function (i.e., the combined statistical significance of all coefficients of the corresponding factor). Effect sizes of continuous factors are represented by partial correlation coefficients ($r_p$), which control for all model covariates and are calculated using the corresponding $t$-statistic and degrees of freedom [54]. The 95% CIs of the effect sizes were derived from the sample variance of the partial correlation [55].

## Results

### Sample characteristics

Children included in our analyses did not appreciably differ in key sociodemographic indicators compared with the entire ABCD cohort (Table 1). Lead-risk scores were bimodally distributed: 40.5% of participants lived in neighborhoods with low lead-risk scores (lead risk ≤ 3; $n$ = 3,450); 32.3%, intermediate lead-risk scores (4 ≤ lead risk ≤ 7; $n$ = 2,749); and 27.3%, high lead-risk scores (lead risk ≥ 8; $n$ = 2,325) (Fig 1A in S1 Appendix). Participants also tended to

**Table 1. Demographics for the Adolescent Brain Cognitive Development (ABCD) study.**

| | Release 2.0.1 (%) | Sample with Complete Data Used in This Study (%) |
|---|---|---|
| **Sex** | | |
| Male | 6,188 (52.1%) | 4,469 (52.4%) |
| Female | 5,681 (47.8%) | 4,055 (47.6%) |
| Missing/Undefined | 6 (0.1%) | 0 (0%) |
| **Income Bracket** | | |
| <$50K (Low) | 3,222 (27.1%) | 2,415 (28.3%) |
| $50-100K (Mid) | 3,070 (25.9%) | 2,482 (29.1%) |
| >$100K (High) | 4,565 (38.4%) | 3,627 (42.6%) |
| Missing/Undefined | 1,018 (8.6%) | 0 (0%) |
| **Race** | | |
| American Indian/Alaska Native | 62 (0.5%) | 42 (0.5%) |
| Asian | 276 (2.3%) | 188 (2.2%) |
| Black | 1,867 (15.7%) | 1,168 (13.7%) |
| Native Hawaiian/Pacific Islander | 16 (0.1%) | 10 (0.1%) |
| Other | 1,958 (16.5%) | 1,364 (16.0%) |
| White | 7,523 (63.4%) | 5,752 (67.5%) |
| Missing/Undefined | 173 (1.5%) | 0 (0%) |
| **Ethnicity** | | |
| Hispanic | 2,409 (20.3%) | 1,640 (19.2%) |
| Not Hispanic | 9,308 (78.4%) | 6,884 (80.8%) |
| Missing/Undefined | 158 (1.3%) | 0 (0%) |
| **Total** | 11,875 | 8,524 |

live in less-deprived neighborhoods (i.e., ≤5th decile; Fig 1B in S1 Appendix). On average, based on housing age, ~20.5% of houses in participants' neighborhoods were estimated to contain lead-based paint hazards (Fig 1C in S1 Appendix) [26,27]. There were positive associations between ADI and both lead risk, $\rho = 0.36$, $p < .001$, and housing age (i.e., estimated percentage of homes in the census tract with lead-based paint hazards based on housing age), $\rho = 0.13$, $p < .001$.

## Lead risk and subcortical brain structure

Of 21 subcortical regions (Fig 1) [56], lead-exposure risk was associated with smaller volumes (mm$^3$) of the posterior ($r_p = -0.022$ [-0.033, -0.011]), mid-posterior ($r_p = -0.035$ [-0.046, -0.024]), central ($r_p = -0.038$ [-0.048, -0.027]), and mid-anterior corpus callosum ($r_p = -0.040$ [-0.051, -0.030]) (Figs 2 and 3). The mid-anterior, central, and mid-posterior corpus callosal associations passed FDR correction ($q < .05$) [49]. Anterior corpus callosum volume was not associated with lead risk ($r_p = -0.004$ [-0.015, 0.007]). No other main effects of lead risk were significant, $Ps \geq .074$ (uncorrected) (Tables 2–22 in S1 Appendix).

For posterior, mid-posterior, central, and mid-anterior corpus callosal volumes, there were no main effects of family income, $F(2, 8508)s \leq 1.10$, $ps \geq .334$ (Tables 13–16 in S1 Appendix), Family Income × Lead Risk interactions (Fig 3), $F(2, 8508)s \leq 0.98$, $ps \geq .376$, or Sex × Lead Risk interactions, $F(1, 8509)s \leq 0.32$, $ps \geq .570$ (Tables 23–26 in S1 Appendix).

## Accounting for neighborhood disadvantage

Correcting for FDR, ADI was inversely associated with subcortical gray-matter ($r_p = -0.037$ [-0.048, -0.026]), cerebellar cortical ($r_p = -0.032$ [-0.043, -0.021]), accumbens area ($r_p = -0.029$

| Subcortical Region | Risk of Lead Exposure | | | Area Deprivation Index | | |
|---|---|---|---|---|---|---|
| | $t(8508)$ | $p$ | $b$ | $t(8508)$ | $p$ | $b$ |
| Corpus Callosum (Mid-Anterior) | -3.73 | < .001 | -1.52 | -0.80 | .421 | -0.48 |
| Corpus Callosum (Central) | -3.47 | .001 | -1.25 | -0.81 | .417 | -0.42 |
| Corpus Callosum (Mid-Posterior) | -3.21 | .001 | -1.03 | -0.80 | .422 | -0.37 |
| Corpus Callosum (Posterior) | -2.03 | .043 | -1.06 | -0.49 | .624 | -0.37 |
| Putamen | -1.78 | .074 | -4.07 | -2.43 | .015 | -8.06 |
| Caudate | -1.60 | .110 | -2.99 | -1.93 | .053 | -5.39 |
| 3rd Ventricle | 1.52 | .129 | 1.34 | -0.34 | .733 | -0.44 |
| Accumbens Area | -1.30 | .193 | -0.43 | -2.72 | .007 | -1.34 |
| Lateral Ventricle | 1.26 | .209 | 13.37 | -0.45 | .652 | -7.09 |
| Subcortical Gray Volume | -1.23 | .220 | -14.77 | -3.39 | .001 | -60.70 |
| Hippocampus | 1.18 | .240 | 1.54 | -1.03 | .303 | -1.91 |
| Inferior Lateral Ventricle | 1.03 | .304 | 0.57 | -0.18 | .860 | -0.15 |
| Ventral Diencephalon | 0.93 | .355 | 1.03 | -1.54 | .123 | -2.57 |
| Pallidum | -0.91 | .365 | -0.63 | -2.20 | .028 | -2.27 |
| Thalamus Proper | -0.41 | .678 | -0.78 | -2.31 | .021 | -6.34 |
| Corpus Callosum (Anterior) | -0.38 | .702 | -0.21 | 1.20 | .231 | 0.94 |
| Cerebellum Cortex | -0.37 | .712 | -6.41 | -2.92 | .003 | -75.54 |
| 4th Ventricle | 0.24 | .812 | 0.55 | -2.17 | .030 | -7.13 |
| Cerebellum White Matter | -0.22 | .829 | -1.59 | -0.43 | .669 | -4.58 |
| Brain Stem | 0.21 | .835 | 1.28 | -0.61 | .544 | -5.55 |
| Amygdala | -0.03 | .977 | -0.02 | -2.69 | .007 | -2.68 |

**Fig 1. Associations of risk of lead exposure and area deprivation index with subcortical volume.** Subcortical regions are sorted vertically by *p*-value (uncorrected) for risk of lead exposure. For each predictor (risk of lead exposure, area deprivation index), the shade of each cell reflects the strength of the association, with redder colors reflecting more positive associations and bluer colors reflecting more negative associations, in accordance with Fig 2. Analyses controlled for age, sex, parental education, race, ethnicity, family income, intracranial volume, and the interaction between family income and either lead risk or ADI. $t(8508)$ = *t*-statistic, 8508 degrees of freedom. $b$ = unstandardized regression coefficient (i.e., change in mm$^3$ regional volume with decile of lead risk or area deprivation index).

[-0.040, -0.019]), and amygdalar volumes ($r_p$ = -0.029 [-0.040, -0.018]) (Figs 1 and 2). In contrast, corpus callosal subregion volumes did not covary with ADI, *ps* ≥ .231 (Tables 27–47 in S1 Appendix). To confirm these dissociations between ADI and lead risk, we reanalyzed these data with the housing-age metric as the predictor of interest. As in the lead-risk analyses, age of housing (i.e., housing-age-based estimates of the proportion of homes with lead-paint hazards) in children's neighborhoods was inversely associated with posterior ($r_p$ = -0.026 [-0.037, -0.015]), mid-posterior ($r_p$ = -0.029 [-0.040, -0.019]), central ($r_p$ = -0.036 [-0.047, -0.025]), and mid-anterior corpus callosal volumes ($r_p$ = -0.034 [-0.045, -0.023]) (Tables 48–68 in S1 Appendix), with the latter three passing FDR correction. Further, when ADI and lead risk (and their interactions with family income) were included in the same model, the FDR-corrected

## Risk of Lead Exposure

## Area Deprivation Index

**Fig 2.** Regional associations between subcortical volume and risk of lead exposure (top) or area deprivation index (ADI) (bottom). For lead risk, these associations, correcting for false-discovery rate (FDR), were significant for mid-posterior, central, and mid-anterior corpus callosum. For ADI, subcortical gray matter, cerebellum cortex, accumbens area (not shown in this image), and amygdala. Regions are color-coded in correspondence to the effect size (i.e., partial correlation coefficient) of lead risk (top) and ADI (bottom), controlling for age, sex, parental education, race, ethnicity, family income, intracranial volume, and the interaction between family income and either lead risk or ADI. Regions with bolded outlines passed FDR correction. Blue-shaded regions indicate inverse associations between lead risk (or ADI) and volume (e.g., greater lead risk, lesser volume), while red-shaded regions indicate positive correlations (e.g., greater lead risk, greater volume). These images were generated in MATLAB using data from the *ggseg* toolbox in R [57]. 3v = 3$^{rd}$ ventricle; 4v = 4$^{th}$ ventricle; Am = amygdala; Bs = brain stem; C = caudate; CCa = anterior corpus callosum; CCc = central corpus callosum; CCma = mid-anterior corpus callosum; CCmp = mid-posterior corpus callosum; CCp = posterior corpus callosum; CeCo = cerebellum cortex; CeWm = cerebellum white matter; H = hippocampus; Lv = lateral ventricle; Pa = pallidum; P = putamen; Th = thalamus; V = ventral diencephalon.

associations between lead risk and mid-anterior ($r_p$ = -0.036 [-0.047, -0.025]), central ($r_p$ = -0.034 [-0.045, -0.023]), and mid-posterior corpus callosal volumes ($r_p$ = -0.030 [-0.040, -0.019]) were maintained ($q$ < .05), as were the FDR-corrected associations between ADI and subcortical gray-matter ($r_p$ = -0.034 [-0.045, -0.023]), amygdalar ($r_p$ = -0.033 [-0.043, -0.022]), and cerebellar cortical volumes ($r_p$ = -0.031 [-0.042, -0.020]) ($q$ < .05) (Tables 117–137 in S1 Appendix). These sensitivity analyses suggest that lead-exposure risk, not income or ADI, was associated with diminished callosal subregion volumes.

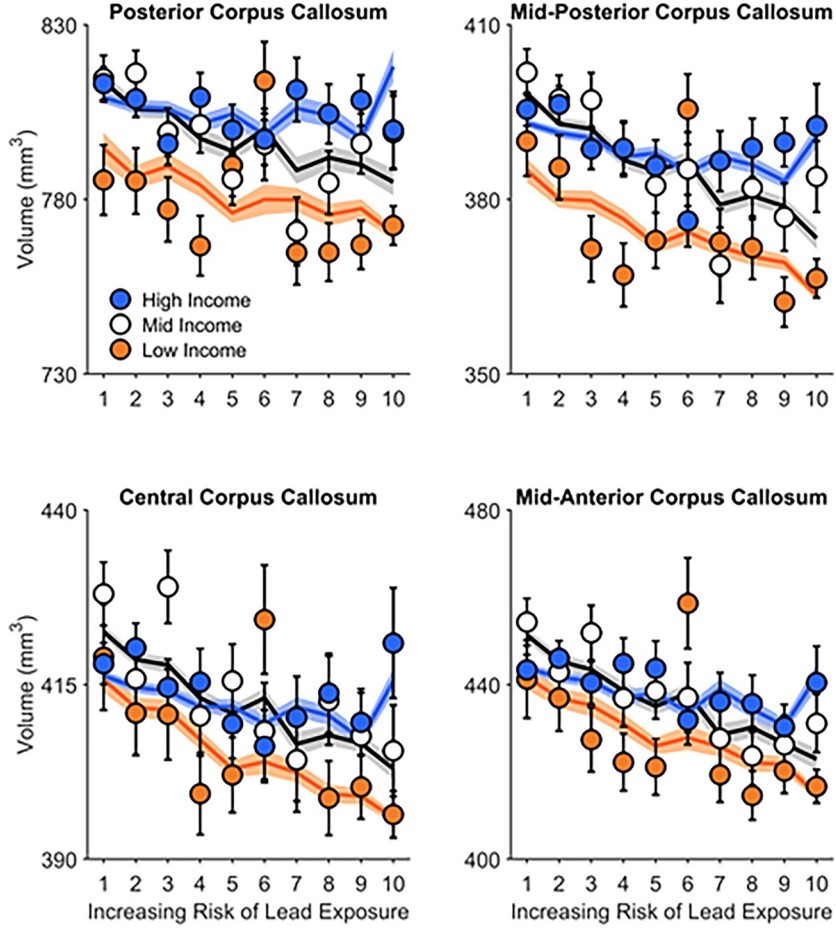

**Fig 3. Posterior, mid-posterior, central, and mid-anterior corpus callosal volume significantly decreased with increasing risk of environmental lead exposure.** These inverse associations were not significantly different between children from low-, mid-, and high-income families. Error bars represent ±1 between-subjects standard error of the observed means. The solid lines represent means of the marginal fitted values of the model; the shaded area surrounding the solid lines represent ±1 between-subjects standard error of those means. Age, sex, parental education, race, ethnicity, and intracranial volume were included as covariates in this analysis.

## Corpus callosum volume and cognition

Next, we analyzed how mid-posterior, central, and mid-anterior corpus callosal volumes were associated with NIH Toolbox performance (Tables 69–103 in S1 Appendix). Mid-posterior corpus callosal volume was positively associated with performance on the Dimensional Change Card Sort Test ($r_p$ = 0.042 [0.031, 0.053]), the Flanker Inhibitory Control and Attention Test ($r_p$ = 0.038 [0.027, 0.049]), the Pattern Comparison Processing Speed Test ($r_p$ = 0.035 [0.024, 0.046]), the Oral Reading Recognition Test ($r_p$ = 0.034 [0.024, 0.045]), and the Picture Vocabulary Test ($r_p$ = 0.056 [0.045, 0.067]). Central corpus callosal volume was positively associated with Dimensional Change Card Sort Test ($r_p$ = 0.027 [0.017, 0.038]) and Picture Vocabulary Test performance ($r_p$ = 0.033 [0.022, 0.044]). Pattern Comparison Processing Speed Test performance was also positively associated with mid-anterior corpus callosal volume ($r_p$ = 0.022 [0.011, 0.033]). Secondary analyses showed that anterior corpus callosal volume was positively associated with performance on the Dimensional Change Card Sort Test ($r_p$ = 0.022 [0.012, 0.033]) and Picture Vocabulary Test ($r_p$ = 0.031 [0.020, 0.042]), and posterior (like

mid-posterior) corpus callosal volume was positively associated with performance on the Dimensional Change Card Sort Test ($r_p$ = 0.045 [0.034, 0.056]), the Flanker Inhibitory Control and Attention Test ($r_p$ = 0.029 [0.019, 0.040]), the Pattern Comparison Processing Speed Test ($r_p$ = 0.032 [0.022, 0.043]), the Oral Reading Recognition Test ($r_p$ = 0.035 [0.024, 0.046]), and the Picture Vocabulary Test ($r_p$ = 0.058 [0.047, 0.069]).

### Neurostructurally-mediated associations between lead risk and cognition

We then examined whether mid-posterior, central, and mid-anterior callosal volumes mediated lead-risk relationships with cognitive performance (Tables 104–116 in S1 Appendix). Our results were consistent with mid-posterior corpus callosal volume mediating associations between lead risk and five cognitive tests (i.e., one test of executive functioning, two of language functioning, one of processing speed, and one of attention/inhibitory control) (Fig 4). Mediation was not present for central or mid-anterior corpus callosal volumes.

## Discussion

Higher neighborhood-level risks of childhood lead exposure were associated with smaller mid-anterior, central, and mid-posterior corpus callosal volumes (i.e., the genu, truncus/body, and anterior splenium, respectively); to a lesser extent, lead-exposure risk was associated with smaller posterior corpus callosal volumes (i.e., posterior splenium) [56,58,59]. These associations were absent for ADI, but present for housing age. Thus, the lead-risk associations were likely attributable to factors associated with elevated lead-exposure risk (e.g., residual lead paint in older homes) [1,27]. Because ABCD enrolled 9- to 10-year-old participants at baseline, we cannot know whether these associations existed prior to baseline (i.e., pre- vs. postnatal insults) or disentangle them from other unmeasured confounding factors. However, the relationships between lead risk and callosal volume in mid-anterior to posterior, but not anterior, callosal regions are consistent with research showing earlier development of anterior than posterior corpus callosum, with the latter continuing to grow through adolescence (i.e., anterior-to-posterior maturation) [60–63]. While we recently reported the strongest inverse associations between lead-exposure risk and cortical structure in children of low-income families [28], the nonspecific inverse associations in callosal structure observed here suggest differential socioeconomic and environmental modulation of cortical and subcortical developmental trajectories, which is consistent with known spatially and temporally variable structural brain maturation [30,64].

Previous research on lead's impact on corpus callosal structure has been inconsistent. Stewart and colleagues [19] reported inverse correlations between bone-lead levels and posterior corpus callosal volume in occupationally exposed lead workers. Brubaker et al. [21] reported increased white matter integrity in the callosal genu, body, and splenium in adults exposed to lead during childhood, but Hsieh et al. [65] found nonsignificant differences in corpus callosal white-matter integrity in lead-exposed workers. Lasky et al. [66] reported no significant differences in callosal volume in lead-exposed (pre- or postnatal) versus non-lead-exposed rhesus monkeys, while Rai et al. [67] suggested that exposure to metal mixtures (arsenic, cadmium, and lead) in developing rats may thin the corpus callosum. There are no endogenous lead-exposure data yet in ABCD, but our study suggests that lead-exposure risk and its related environmental factors may be associated with corpus callosal morphology.

Exposures to other neurotoxicants are similarly associated with callosal morphology [68–72]. Prenatal alcohol exposure was most consistently associated with smaller areas of more posterior callosal regions in 8- to 22-year-olds [68,69], and prenatal particulate matter air pollution exposure, especially during the 3rd trimester, was most strongly associated with smaller

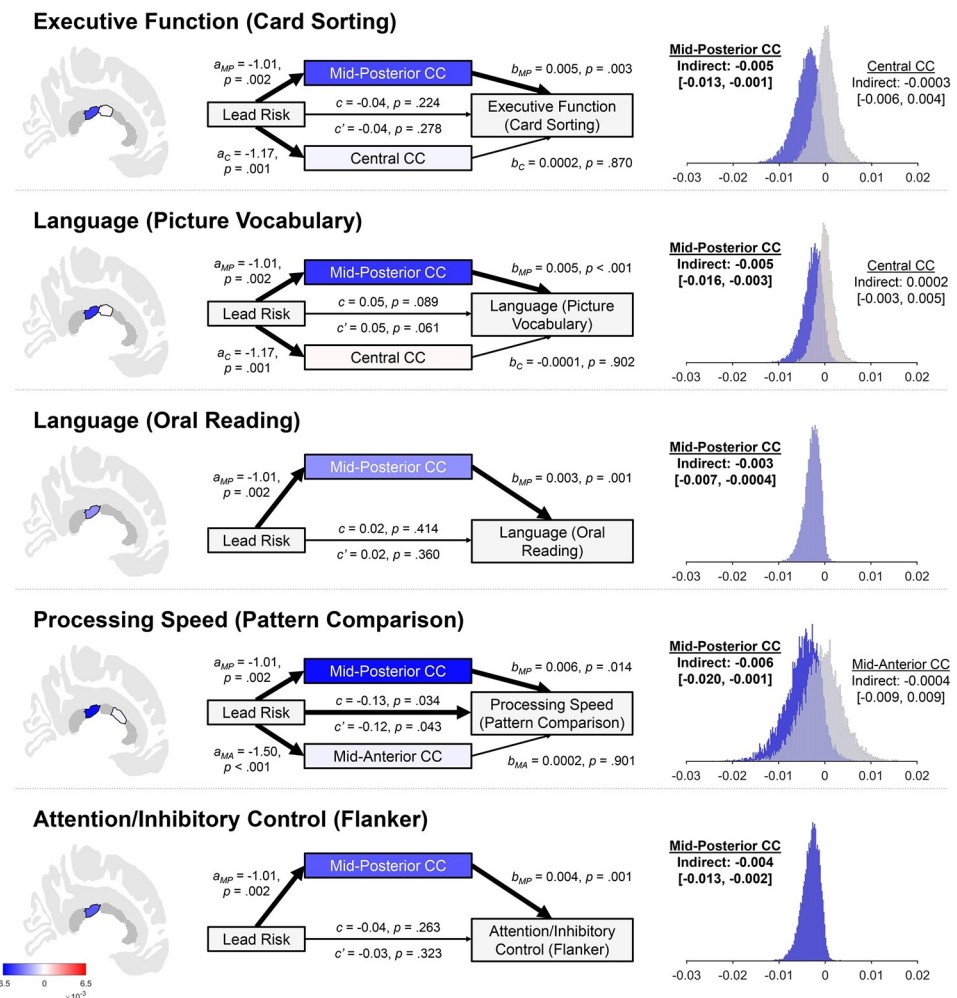

**Fig 4. Mediational analyses showing indirect associations of lead risk on cognitive performance through mid-posterior corpus callosal (CC) volume.** The title of each row [e.g., "Executive Function (Card Sorting)"] refers to the construct and task used to assess that construct. The shading of the callosal regions, the mediator boxes (e.g., Mid-Posterior CC, Central CC), and histograms reflect the strength of the indirect association (per the color bar in the lower left corner). Callosal regions in dark gray were not incorporated as mediators in the corresponding analysis. Total effects of lead risk are represented by *c*, direct effects of lead risk are represented by *c'*, and *a* and *b* values refer to the associations of lead risk on callosal volume and callosal volume on cognitive performance, respectively. The subscript of the *a* and *b* values refer to the callosal subregion (i.e., *C* = central, *MA* = mid-anterior, *MP* = mid-posterior). All *a*, *b*, *c*, and *c'* values are unstandardized regression coefficients. Thick arrows designate significant associations. Thin arrows designate non-significant associations. The presence of the statistical significance of indirect associations was determined by construction of bias-corrected percentile bootstrapping 95% confidence intervals (i.e., 10,000 bootstrapped samples), irrespective of the statistical significance of the corresponding total effect. The distributions of these bootstrapped indirect associations are shown, along with the original-sample indirect associations and corresponding confidence intervals. Bolded font indicates statistically significant indirect associations. Age, sex, parental education, race, ethnicity, and family income were included as covariates in analyses. Card Sorting = Dimensional Change Card Sort Task. Picture Vocabulary = Picture Vocabulary Test. Oral Reading = Oral Reading Recognition Test. Pattern Comparison = Pattern Comparison Processing Speed Test. Flanker = Flanker Inhibitory Control and Attention Test.

volumes of the callosal body in 8- to 12-year-olds (mid-anterior + central + mid-posterior) [70]. Posterior corpus callosal size was 3.1% (7.9%) smaller in 12- to 18-year-old males (females) prenatally exposed to cigarette smoking [71], and splenial (posterior corpus callosal) size in 4.5-year-olds was 1.4% smaller given high versus low levels of prenatal polychlorinated-

biphenyl exposure [72]. Here, we found a 5.2% mean decrease in mid-posterior, central, and mid-anterior corpus callosal volume in individuals living in census tracts with the highest versus lowest lead-risk score.

Lead's well-established effects on cognition and potential associations with corpus callosal structure are complemented by relationships between corpus callosal structure, cognition, and intelligence [73]. In our sample, anterior callosal regions were weakly associated with cognitive function whereas posterior regions were more strongly associated with cognitive performance. The association between mid-posterior callosal structure and processing speed (i.e., the Pattern Comparison Processing Speed Test) corroborates research on callosal volume and processing speed in occupationally lead-exposed and non-exposed adult men [18]. Our reported associations between mid-posterior callosal volume and performances on the Picture Vocabulary Test and the Oral Reading Recognition Test are also consistent with research suggesting that posterior callosal regions are critical for interhemispheric transfer between temporal-parietal-occipital cortical regions involved in language processing [74–77].

The lead-risk metric was primarily a function of the estimated neighborhood-level prevalence of lead-based paint given the age of houses in that neighborhood [25,26]. Even though lead-based house paint and leaded gasoline (for on-road vehicles) were banned in the US in 1978 [78] and 1996 [79], children remain at risk for ingesting lead via (1) drinking water provided through lead service lines (i.e., lead-containing plumbing) [80], (2) lead-contaminated dust and soil given the prior use of lead-based paint in older buildings [27], and (3) lead-contaminated topsoil from past leaded-gasoline vehicle emissions [81]. Indeed, while average blood lead levels have substantially declined over the past several decades, a 2021 study estimated that nearly 400,000 1-to-11-year-olds in 2011–2016 had blood-lead levels exceeding the CDC's reference level of 5 μg/dL [82]. Here, about one in five houses in our participants' neighborhoods were estimated to contain lead-based paint hazards, which is consistent with national surveys showing that 25% of United States housing stock contains one or more lead hazards [27]. Of 14 risk factors known to impair neurodevelopment (e.g., medical conditions, low SES), lead exposure was reported as 2nd only to preterm birth in its impact on total reduction in IQ [83]. Lead-associated decrements in IQ have been suggested to contribute to annual costs of approximately $977 billion in low- and middle-income countries [17], many of which do not have regulations as to limits of the concentration of lead in paint [84]. Accordingly, reducing risks of lead exposure (and addressing related environmental factors) will have economic and health benefits that facilitate prosperity in the United States and worldwide.

Lead risk and housing age were distinctly associated with callosal morphology. In contrast, ADI, a metric of neighborhood socioeconomic disadvantage [40,41], was inversely associated with subcortical gray-matter, cerebellar cortical, accumbens area, and amygdalar volumes, but not callosal morphology. Despite research on the effects of neighborhood SES on child development (e.g. ADI) [85] and family-specific SES measures on children's brain structure [86–88], less is known about how neighborhood-level metrics of disadvantage influence neurocognitive development [87,89]. Our results are consistent with previous reports showing (1) smaller cortical and subcortical volumes in 8- to 21-year-olds living in low-SES neighborhoods [90] and (2) greater age-related increases in right amygdalar volume in adolescents living in more disadvantaged neighborhoods (potentially due to the groups' lower baseline amygdalar volumes) [91]. Given associations between amygdalar volume and depressive symptoms [92], it will be critical to evaluate longitudinal trajectories of mental health and brain development in ABCD and whether neighborhood characteristics increase risk or promote resilience to developmental insults such as lead neurotoxicity.

## Limitations and future directions

In the current report, the primary residential addresses of 9- to 10-year-old children were used to derive community-based risk estimates of lead exposure, which our past research has shown are valid proxies of exposure [28]. Peak lead exposure during childhood occurs in toddlers (i.e., 2- to 3-years-old) from the confluence of hand-to-mouth ingestion of lead-contaminated floor dust, soil, water, and paint chips [93], but older children are also vulnerable to lead toxicity [28,94,95]. For example, IQ in older children has been shown to be better predicted by concurrent than past blood-lead levels [94,96], and, even at low levels of exposure, IQ was shown to be associated with concurrent blood-lead levels in 7- to 14-year-olds [97,98]. Ultimately, the age of greatest vulnerability to lead neurotoxicity is unclear [95], but recent evidence has suggested that exposure to other neurotoxicants (i.e., air pollution) is also associated with brain structure in 9- to 10-year-olds [99].

As we cannot physically manipulate lead exposure in ABCD participants, the longitudinal design of the ABCD Study and its diverse cohort offer opportunities to evaluate how developmental trajectories of brain and cognitive development are associated with differential risks of lead exposure. The ABCD Study does not yet have endogenous lead-exposure data in its participants, but it has been actively collecting address histories for its participants since birth, which, when completed, will facilitate understanding of the critical developmental periods of lead neurotoxicity vulnerability. Similarly, even though ABCD's data collection sites are primarily in metropolitan areas, the recruitment areas of these 21 sites represent at least 20% of the 9-to-10-year-old US population [35]. Further, while the potential for reverse causation is inherent to cross-sectional studies, thereby limiting causal inference, it is unlikely that poor cognitive performance elicits altered brain structure, or that altered brain structure induces risk factors of lead exposure here, thus supporting the temporal ordering within our cross-sectional mediational analyses (lead risk → brain structure → cognitive performance) [100].

While ADI and lead risk were strongly correlated, over 85% of the variance in lead risk was not accounted for by ADI, reflecting a possible necessity to analyze multiple factors of environmental health disparities in future research [101]. Specifically, our analyses showed dissociations between the subcortical regions associated with lead risk and ADI, and the associations between lead risk and corpus callosal volumes were maintained when including ADI in another set of models. Thus, our results suggest that subcortical brain structure (i.e., corpus callosal volume) in adolescents may be uniquely associated with factors related to increased risk of lead exposure (i.e., age of housing, with older homes more likely to contain lead-based paint hazards) [27].

Even though the ADI and lead-risk data reflect community- rather than individual-level estimates, past research has argued that using community-level data to evaluate environmental risks may ultimately prevent exposure to the hazards before the individual is actually exposed to them (i.e., screening communities/homes before occupancy) [1,102]. Indeed, individual screening questionnaires of potential lead exposure may not accurately identify the children with elevated blood lead levels [103]. Therefore, while community-level risk estimates inherently over- and underestimate risks of specific individuals in that community, future incorporation of such geocoded data in cognitive neuroscience research may considerably advance understanding of the environmental contextualization of an individual's neurocognitive and brain development, especially given the more common practice of collecting data pertaining to the individual (e.g., SES).

As individuals are rarely exposed to isolated chemicals (but to mixtures of chemicals) [104], the incorporation of multiple data sources reflecting "mixtures" of environmental health disparities may also offer substantial insight into the collective and synergistic factors to target in

environmental remediation interventions. Simply, the risk of lead exposure does not exist in a vacuum but is associated with past and current practices that have differentially subjected children to such risks. For example, the burden of lead-exposure's effects is typically greatest in children in the lowest SES families [105–108], a glaring example of environmental injustice [105,109]. Similarly, Black and Hispanic children tend to have greater mean blood lead levels than white children [5,110–112] and are more likely than white children to live in homes or regions with greater risks of lead exposure [113–115]. Further, lead-poisoning rates (and, thus, children's blood-lead levels) are associated with multiple community-level factors [39], including value and age of houses, poverty rates, population density, and percentage of the population who are Black or Hispanic [1,116], signifying racial residential segregation as a potential explanatory mechanism for lead exposure disparities [117]. While the data in the current manuscript may reflect differences in lead exposure, these differences would then ultimately be due to disparate conditions that initially elicited such differences, thereby focusing any potential intervention efforts on the originating disparities. Indeed, recent research has shown that soil-lead concentrations tended to be elevated in samples taken from historically redlined neighborhoods compared to those in "best" and "desirable" neighborhoods, per zone designations by the 1930's Home Owners' Loan Corporation [118]. Ultimately, because we do not currently have endogenous lead-exposure data in our participants, the results related to risk of lead exposure may be alternatively explained (at least partially) by these other systemic and environmental factors. Accordingly, upon collection of bodily lead data and additional geocoded data in ABCD, our future research will involve analyses of both chemical and "environmental-disparity mixtures" to both study developmental trajectories more comprehensively and evaluate the relative strengths of the associations between adolescent development and other sources of disparity (e.g., air pollution, residential segregation).

## Conclusion

Lead-induced cognitive deficits are likely governed by how lead exposure influences brain structure [18], and our results, consistent with callosal mediation of lead-risk associations with cognition, offer the testable hypothesis in future studies that "dose" of lead exposure mediates cognitive functioning through changes in mid-posterior corpus callosal structure. ABCD is measuring lead concentrations in shed deciduous (baby) teeth [119–121], but those data will not be available for several years. ABCD is also exploring the collection and analysis of blood lead levels to gauge how well they are correlated with neighborhood-level lead-risk estimates. ABCD is also working to incorporate electronic health records as part of its dataset, which may also help elucidate these relationships via past lead screening results. Until then, this study, which uses neighborhood-level lead-exposure risk, provides potential evidence that cognitive deficits from low-level lead toxicity (and its related environmental factors) may operate by diminishing subcortical brain structure [28].

## Supporting information

**S1 Appendix. Supplementary figures and statistical output tables.**
(DOCX)

## Acknowledgments

*ABCD Acknowledgement*. Data used in the preparation of this article were obtained from the Adolescent Brain Cognitive Development (ABCD) Study (https://abcdstudy.org/), held in the NIMH Data Archive (NDA). This is a multisite, longitudinal study designed to recruit more

than 10,000 children aged 9–10 and follow them over 10 years into early adulthood. A full list of supporters is available at https://abcdstudy.org/federal-partners/. A listing of participating sites and a complete listing of the study investigators can be found at https://abcdstudy.org/principal-investigators/. ABCD consortium investigators designed and implemented the study and/or provided data but did not necessarily participate in analysis or writing of this report. This manuscript reflects the views of the authors and may not reflect the opinions or views of the NIH or other ABCD consortium investigators. The ABCD data repository grows and changes over time. The ABCD data used in this report came from https://doi.org/10.15154/1523347.

## Author Contributions

**Conceptualization:** Andrew T. Marshall, Elizabeth R. Sowell.

**Formal analysis:** Andrew T. Marshall, Wesley K. Thompson.

**Visualization:** Andrew T. Marshall.

**Writing – original draft:** Andrew T. Marshall, Rob McConnell, Bruce P. Lanphear, Wesley K. Thompson, Megan M. Herting, Elizabeth R. Sowell.

**Writing – review & editing:** Andrew T. Marshall, Rob McConnell, Bruce P. Lanphear, Wesley K. Thompson, Megan M. Herting, Elizabeth R. Sowell.

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
