## [Decision Letter · Decision Letter 0]

20 Jan 2021

PONE-D-20-37575

Risk of lead exposure, subcortical brain structure, and cognition in a large cohort of 9-10-year-old children

PLOS ONE

Dear Dr. Marshall,

We were pleased to receive your manuscript at PLOS ONE. After careful consideration, we feel that it has merit but does not fully meet PLOS ONE’s publication criteria as it currently stands. Therefore, we invite you to submit a revised version of the manuscript that addresses the points raised during the review process. Lead exposure is an important area of concern for development and in my opinion continues to be a major pediatric problem.

The authors should respond to all of the points raised by the reviewer and myself and include/insert the additional data requested. There are a number of questions about the relation between the lead risk and ADI variables, the ICV, and enumeration of additional limitations, e.g., the inability to determine whether pre- or postnatal lead exposure is involved, that should be added to the end of the Discussion. 

We look forward to receiving your revised manuscript.

Kind regards,

Sandra W. Jacobson, Ph.D.

Academic Editor

PLOS ONE

Journal Requirements:

Additional Editor Comments:

Using data from the Adolescent Brain Cognitive Development (ABCD) Study, the authors recently reported decreases in cortical surface area and volume with increased risks of lead exposure, primarily in children of low-income families. They used data from 8,524 typically developing 9- and 10-year-old children and their parents/caregivers from the ABCD Study, who had originally been recruited from 22 study sites. The primary outcomes and measures studied in the present paper are subcortical brain structure, cognitive performance using the National Institutes of Health Toolbox, and geocoded risk of lead exposure. The authors reported that children from neighborhoods with greater risks of environmental lead exposure exhibited smaller volumes of the mid-anterior (partial correlation coefficient, central, and mid-posterior corpus callosum. Smaller volumes of these three callosal regions were associated with poorer performance on language and processing speed tests. The association of lead exposure risk on cognitive performance was partially mediated through callosal volume, particularly the mid-posterior corpus callosum. In contrast, neighborhood-level indicators of disadvantage were not associated with smaller volumes of these brain structures.

The paper is well-written, although sometimes a little too terse, and more information would be helpful for the reader. The question of lead exposure is important, and the problem continues to affect many children and families. The authors provide an up-to-date lit review and include a major lead researcher among the authors (BL). It is, therefore, surprising that the pediatric problem studied in this paper was not considered sufficiently prevalent for publication in PLOS Medicine—their loss—and an opportunity for PLOS ONE to publish findings about this major exposure and its subsequent consequences.

The authors refer to secondary analyses in which ADI replaces lead risk. What is the association between the lead risk score and ADI? What happens if the authors include both the lead risk score and ADI in the same analyses instead of replacing lead risk with ADI?

The findings in Table 2 indicate larger ventricles in the presence of greater lead—the need to control for smaller ICV is important and should probably be included in the findings presented in Table and in the text.

The major limitation of the paper is the potential overlap between lead risk including housing age vs. ADI, a metric of neighborhood SES disadvantage. Although the authors attempt to discriminate between them, this is not entirely convincing. Another limitation that should be repeated in the Conclusions is that these findings are based on a community-related risk rather than on individual risk.

Lines 130+ and later 314+ and wherever referring to these tests: The names of the cognitive tests used should be cited properly by specifying name of test and capitalizing words appropriately (e.g., Picture Vocabulary Test; Flanker test, etc.)

Line 185 Why wasn’t ICV included as a covariate? Figure 1 indicates that ICV was a covariate. There appears to be an inconsistency here—please explain or correct.

Title and in lines 364 and elsewhere in Discussion, where age range is given: change “9-10-year-old” to “9- to 10-year-old”

Lines 440-441 needs to add not only “how” but “when” lead exposure occurs. The authors acknowledge that they could not distinguish between pre- and postnatal exposure and thus timing of exposure should be repeated here.

The authors should respond to all of these comments and questions as well as those raised by the reviewer, who also commented on the questions of pre- vs. postnatal lead exposure, the ICV, raised questions about criteria used to select confounders and adding statistical data to Table 1.

Reviewers' comments:

Reviewer's Responses to Questions

**Comments to the Author**

1. Is the manuscript technically sound, and do the data support the conclusions?

Reviewer #1: Yes

2. Has the statistical analysis been performed appropriately and rigorously? 

Reviewer #1: Yes

3. Have the authors made all data underlying the findings in their manuscript fully available?

Reviewer #1: Yes

4. Is the manuscript presented in an intelligible fashion and written in standard English?

Reviewer #1: Yes

5. Review Comments to the Author

Reviewer #1: This study examined associations of lead exposure risk with regional subcortical brain volumes and cognitive test performance, and examined whether the association between lead risk and cognitive performance was mediated by subcortical volumes. The study cohort formed part of a multi-site study of typically developing adolescents. Children with higher risk of lead exposure were observed to have smaller volumes of three callosal regions and poorer performance on cognitive tests relating to language and processing speed. Furthermore, the volume of the mid-posterior corpus callosum was shown to partially mediate the association of lead exposure risk with cognitive performance. These findings agree with previous studies showing that high levels of lead exposure are associated with changes in brain structure and volume and extend them to show similar effects of lower levels of exposure in children. This is an interesting, well-designed and well-written study, and given that lead exposure is a significant risk factor for neurodevelopmental impairment, it provides an important contribution to this field.

Abstract:

Page 7, line 52: “The association of lead exposure risk on cognitive performance…” should rather read “The association of lead exposure risk with cognitive performance…”

Introduction:

The previous literature was adequately discussed, as well as the rationale for the current study.

Methods:

Page 11, line 111: “Ethnical” should be corrected to “ethical”.

It would be helpful for the reader for the authors to give a summary of the data collection methods (NIH toolbox, and MRI acquisition and processing), even though these have been described elsewhere.

Since this was a multisite study, it would be helpful to know how comparable the data (both in terms of acquisition and outcome measurements) was across the different sites, particularly given that the primary lead exposure risk measure was, as I understand it, dependent to some extent on geographical location. Were the exposure risk metrics similarly distributed across the cohorts at the different sites?

What was the rationale behind the use of the specific confounders included in the analyses? Were these confirmed in the current cohort to be associated with lead exposure risk and/or the outcomes used?

The authors state that they controlled for ICV. Was this associated with lead exposure risk in this cohort?

Results:

Table 1: the authors state that the children in the analyses did not differ appreciably in key sociodemographic indicators compared to the entire ABCD cohort (page 16, lines 205-206). Was this determined by statistically comparing the two? If so the statistical results should be presented in the table for each indicator.

Figures: color-coding the regions (also in Table 2) to indicate the direction and size of the effect is helpful.

The findings of differential effects of neighbourhood ses and neighbourhood lead exposure risk on subcortical morphology is interesting, and the analytical approach was clearly demonstrated here.

Discussion:

The findings are placed in the context of the previous literature, both of lead exposure and of exposure to other neurotoxic substances, and their implications are clearly discussed. Some study limitations are addressed although the possibility of other toxic exposures known to affect brain volume (particularly pre-natally) is not specifically mentioned and should be addressed.

6. PLOS authors have the option to publish the peer review history of their article (what does this mean?). If published, this will include your full peer review and any attached files.

Reviewer #1: No

---

## [Author Response · Author response to Decision Letter 0]

3 Feb 2021

Additional Editor (E) Comments:

1. The paper is well-written, although sometimes a little too terse, and more information would be helpful for the reader.

a. Thank you to the editor for this suggestion. A series of details have been added throughout the manuscript, as described throughout the rest of the responses below.

2. The authors refer to secondary analyses in which ADI replaces lead risk. What is the association between the lead risk score and ADI? What happens if the authors include both the lead risk score and ADI in the same analyses instead of replacing lead risk with ADI?

a. We have added a sentence to the Results section noting the correlation between ADI and both lead risk and housing age. We have also respecified the ADI and housing-age analyses as “sensitivity” analyses, rather than “secondary” analyses. Additionally, per this recommendation, we re-ran the regression analyses with both ADI*Family Income and Lead Risk*Family Income interactions in the same model. The significant associations between lead risk and mid-anterior, central, and mid-posterior corpus callosal volumes continued to pass FDR correction. Accordingly, the following was added to the “Accounting for Neighborhood Disadvantage” section: “Further, when ADI and lead risk (and their interactions with family income) were included in the same model, the FDR-corrected associations between lead risk and mid-anterior (rp = 0.036 [ 0.047, 0.025]), central (rp = 0.034 [ 0.045, 0.023]), and mid-posterior corpus callosal volumes (rp = 0.030 [ 0.040, 0.019]) were maintained (q < .05), as were the FDR-corrected associations between ADI and subcortical gray-matter (rp = 0.034 [ 0.045, 0.023]), amygdalar (rp = 0.033 [ 0.043, 0.022]), and cerebellar cortical volumes (rp = 0.031 [ 0.042, 0.020]) (q < .05) (Tables 117-137 in S1 Appendix” (p. 18, Lines 412-419). Accordingly, the corresponding statistical output tables have been added to Appendix S1 of the supporting information.

3. The findings in Table 2 indicate larger ventricles in the presence of greater lead—the need to control for smaller ICV is important and should probably be included in the findings presented in Table and in the text.

a. We’d like to thank the editor for noting the importance of ICV in these analyses. As specified in the Methods sections (pp. 8-10), the analyses in which the criterion was subcortical volume included ICV as a covariate, even for the non-significant associations of lead risk or ADI (such as the associations with ventricular volume). For the reader’s benefit, we have added the following sentence to the Table 2 caption: “Analyses controlled for age, sex, parental education, race, ethnicity, family income, intracranial volume, and the interaction between family income and either lead risk or ADI.”

4. The major limitation of the paper is the potential overlap between lead risk including housing age vs. ADI, a metric of neighborhood SES disadvantage. Although the authors attempt to discriminate between them, this is not entirely convincing. Another limitation that should be repeated in the Conclusions is that these findings are based on a community-related risk rather than on individual risk.

a. A paragraph has been added to the newly included “Limitations and Future Directions” section describing relationships between ADI, lead risk, and housing age, as well as community-level versus individual-level risk (pp. 25-27).

5. Lines 130+ and later 314+ and wherever referring to these tests: The names of the cognitive tests used should be cited properly by specifying name of test and capitalizing words appropriately (e.g., Picture Vocabulary Test; Flanker test, etc.)

a. This edit has been made throughout the manuscript, and we are appreciative of the editor’s recommendation.

6. Line 185 Why wasn’t ICV included as a covariate? Figure 1 indicates that ICV was a covariate. There appears to be an inconsistency here—please explain or correct.

a. We thank the editor for raising this point. In the first set of analyses of lead risk and brain volume, in order to determine which regional volumes were associated with lead risk (i.e., in which the criterion for analysis was regional volume), we controlled for ICV, allowing us to separate whether greater lead risk was uniquely associated with subcortical volumes. Not including ICV in these analyses would have basically prevented us from knowing if lead risk was associated with smaller regional volumes or just smaller overall brain size. However, in these subsequent analyses, the ultimate criterion was cognitive test score, in that we tested whether the association between lead risk and cognitive test score was mediated by subcortical volume. In the total and direct effect testing of the association between lead risk and cognitive score, we did not control for ICV, in accordance with our past research. Thus, in order to accurately conduct mediation analyses in which the models to measure the indirect effect only included the additional mediator variable rather than a mediator variable and a new covariate (i.e., ICV), we did not control for ICV in these analyses. We have added a sentence to page 10: “Accordingly, because these analyses ultimately tested whether the association between lead risk and cognitive test score was mediated by subcortical volume, in which the criterion was cognitive test score and not subcortical volume, ICV was not included as a covariate for these analyses of cognitive performance.”

7. Title and in lines 364 and elsewhere in Discussion, where age range is given: change “9-10-year-old” to “9- to 10-year-old”.

a. This edit has been made in title and throughout the discussion in accordance with the editor’s recommendation. 

8. Lines 440-441 needs to add not only “how” but “when” lead exposure occurs. The authors acknowledge that they could not distinguish between pre- and postnatal exposure and thus timing of exposure should be repeated here.

a. On pages 25-27, we have divided the final paragraphs into “Limitations & Future Directions” and “Conclusion” sections. The “Limitations and Future Directions” section has also been expanded upon and now includes discussion of the “how” and “when” of lead exposure, per the editor’s request (pp. 25-27)

Comments to the Author: Reviewer #1 (R1)

1. Abstract: Page 7, line 52: “The association of lead exposure risk on cognitive performance…” should rather read “The association of lead exposure risk with cognitive performance…”

a. This sentence was edited in accordance with the reviewer’s recommendation (Abstract).

2. Methods, Page 11, line 111: “Ethnical” should be corrected to “ethical”.

a. This word was corrected in accordance with the reviewer’s correction (p. 6, Line 113).

3. It would be helpful for the reader for the authors to give a summary of the data collection methods (NIH toolbox, and MRI acquisition and processing), even though these have been described elsewhere.

a. Additional details have been added about the specific versions used for the NIH Toolbox, and a general description of cross-task data collection methods for the NIH Toolbox has been added (pp. 7-8). Additional details have also been provided for the MRI acquisition and processing (pp. 7-8). We also provided an additional link to an ABCD-Study-provided PDF that outlines all the neuroimaging parameters across the scanners used at different sites.

4. Since this was a multisite study, it would be helpful to know how comparable the data (both in terms of acquisition and outcome measurements) was across the different sites, particularly given that the primary lead exposure risk measure was, as I understand it, dependent to some extent on geographical location. Were the exposure risk metrics similarly distributed across the cohorts at the different sites?

a. We have added specification that all data-collection sites follow similar experimental protocols, with an added citation of Auchter et al. (2018) from a special issue of Developmental Cognitive Neuroscience on the ABCD Study. The reviewer also noted geographic differences in lead-risk scores. Because we have previously published site-by-site distributions, which we specified in the manuscript, we have added the following sentence to the Methods section: “Briefly, for some sites, the lead-risk scores were uniformly distributed, while other sites showed modalities at either lower, intermediate, and/or higher lead-risk scores, thereby reflecting geographic differences in risk of exposure [39]” (p. 6, Lines 121-124).

5. What was the rationale behind the use of the specific confounders included in the analyses? Were these confirmed in the current cohort to be associated with lead exposure risk and/or the outcomes used?

a. We have included the following sentence in the Methods section addressing the rationale for including the specific covariates used in analyses: “In accordance with previous research analyzing associations between brain structure and socioeconomic and/or environmental conditions [28, 45, 46], we controlled for children’s age, sex, race, ethnicity, maximum parental education, and family income.” (p. 8, Lines 255-256)

6. The authors state that they controlled for ICV. Was this associated with lead exposure risk in this cohort?

a. The primary reason for controlling for ICV initially was for the sake of the subcortical volume analyses. Specifically, it was important to know whether any differences in subcortical volume were due to differences in brain size (here, operationally defined as ICV). However, we thank the reviewer for the suggestion to look at how ICV was correlated with lead risk. Accordingly, in our Methods section, we included the following: “Analyses of subcortical data also controlled for intracranial volume (ICV), which was inversely associated with both lead risk, Spearman’s rho (ρ) = 0.11, p < .001, and ADI, ρ = 0.11, p < .001.” (p. 8, Lines 258-260)

7. Results, Table 1: the authors state that the children in the analyses did not differ appreciably in key sociodemographic indicators compared to the entire ABCD cohort (page 16, lines 205-206). Was this determined by statistically comparing the two? If so the statistical results should be presented in the table for each indicator.

a. We thank the reviewer for this comment and the careful reading/review of our manuscript. In our manuscript, our statement that the demographic indicators in the entire cohort vs. the analyzed sample “did not differ appreciably” was based on observed comparisons between the proportions of participants of each demographic within each factor (e.g., sex, ethnicity). Traditionally, comparing the distribution of participants across different levels of a factor is done through chi-squared goodness-of-fit tests, which works particularly well for samples that are smaller than the sample analyzed here. More specifically, chi-squared tests are very sensitive to sample size, in that, with larger and larger sample sizes, it is very easy to find statistically significant distributional differences across a factor even though such differences, proportionately, are not meaningful. For example, when running chi-squared tests on the demographic indicators in Table 1, the tests for all factors (except sex) are statistically significant, indicating that the observed counts (i.e., the “Sample with Complete Data” column in Table 1) substantially differ from predicted counts (i.e., if 8,524 participants were proportioned in accordance with the percentages data in the “Release 2.0.1” column). However, the percentages (%) / proportion data tell a different story, as they are quite similar between the two columns. 

To illustrate via worked-out example using the ethnicity data, there were 1,640 Hispanic participants and 6,884 non-Hispanic participants in the analyzed sample (i.e., the observed counts). 20.3% of ABCD is Hispanic, while 78.4% is not Hispanic. However, of the participants with non-missing data for ethnicity, 20.6% are Hispanic; 79.4% are not Hispanic. If that same distribution across ABCD was present in the analyzed sample, then we would predict there to be 1,753 Hispanic participants and 6,772 non-Hispanic participants (i.e., the predicted counts). Chi-squared tests would then reveal significant differences between the observed and predicted counts, p = .003. However, the differences with respect to proportion Hispanic versus not Hispanic are quite similar. Indeed, if we were to scale the counts by a factor of 100 (i.e., all counts divided by 100) and re-run the chi-squared tests, then the observed and predicted counts do not differ, p = .763. I contrast, if we were to multiply all of the original counts by 100, then there would be even greater differences between the observed and predicted counts, p < 10e-200. Thus, despite equivalent proportional distributions in all cases, larger samples are vulnerable to statistically significant differences that are not meaningful. Further, after additional research and review of the literature, we were unable to find any sample-size corrections for chi-squared tests for large samples (i.e., instead, many statistical tests instead have corrections for small samples). Accordingly, we believe that including statistical comparisons (e.g., chi-squared tests) for Table 1 would not be informative to the reader and may even cause greater confusion, as they would not reflect the actual proportional distributions, which more effectively capture demographical distribution. Therefore, we have maintained our original statement in the manuscript, “Children included in our analyses did not appreciably differ in key sociodemographic indicators compared with the entire ABCD cohort (Table 1).” (p. 11, Lines 310-311).

8. Discussion: The findings are placed in the context of the previous literature, both of lead exposure and of exposure to other neurotoxic substances, and their implications are clearly discussed. Some study limitations are addressed although the possibility of other toxic exposures known to affect brain volume (particularly pre-natally) is not specifically mentioned and should be addressed.

a. As described above, we have split up our discussion to now have a “Limitations and Future Directions” section (pp. 25-27). From our initial submission, our manuscript includes a paragraph discussing how exposure to other neurotoxicants is associated with brain structure. This passage includes exposure that occurs both pre- and postnatally. In this revision, we have substantially expanded on our discussion of limitations/future directions, including passages on exposures to other neurotoxicants and environments that may synergistically influence brain and cognitive development.

---

## [Decision Letter · Decision Letter 1]

12 Apr 2021

PONE-D-20-37575R1

Risk of lead exposure, subcortical brain structure, and cognition in a large cohort of 9- to 10-year-old children

PLOS ONE

Dear Dr. Marshall,

Thank you for submitting your manuscript to PLOS ONE. After careful consideration, we feel that it has merit but continues to need more work in order to address questions and comments raised in the reviews. Therefore, we invite you to submit a revised version of the manuscript that addresses the points raised during the review process.

There continue to be important questions raised about the resubmission, including questions raised by reviewer 2 (see below) regarding whether there is a direct measure of blood or urine lead that can be added to the manuscript. This would strengthen the conclusion that it is lead and not other aspects of the poor housing that are actually responsible for the outcomes reported. The relations between ADI and both lead risk and housing age that I inquired about were apparently not so strong as to suggest that the ADI and lead risk are one and the same and suggest that they were making relatively independent contributions. However, these analyses were based on lead “risk” and not a direct measure of lead exposure. This point should be made clear in the Discussion. 

We look forward to receiving your revised manuscript.

Kind regards,

Sandra W Jacobson

Academic Editor

PLOS ONE

Additional Editor Comments (if provided):

The authors have provided thoughtful responses in their resubmission of their manuscript on the risk of lead effects, subcortical brain structure, and cognition in school-age children from the large ABCD study. Their detailed responses regarding the statistical analyses helped clarify their approach and provided support for some of the findings. For example, they elaborated on the role of the mediation of the effects of lead exposure and disadvantaged demographic environment on cognition, while also taking into account when to adjust statistically for TIV. Moreover, the clarity with which they provided these responses--comparing alternative data that would have emerged using different approaches–was impressive.

However, there continue to be important questions raised about the resubmission, as indicated by the questions and comments by the additional reviewer, particularly regarding whether there is a direct measure of blood or urine lead, even for a subset of the ABCD sample, available. This would strengthen the conclusion that it is lead and not other aspects of the poor housing that are actually responsible for the outcomes reported here. The relations between ADI and both lead risk and housing age that I inquired about were apparently not so strong as to suggest that the ADI and lead risk are one and the same and suggest that they were making relatively independent contributions. However, these analyses were based on lead “risk” and not a direct measure of lead exposure. As noted by the second reviewer, concluding that lead is the cause of the effects is overstated. Please respond to all of the points raised by this reviewer and modify the manuscript accordingly.

Reviewers' comments:

Reviewer's Responses to Questions

**Comments to the Author**

1. If the authors have adequately addressed your comments raised in a previous round of review and you feel that this manuscript is now acceptable for publication, you may indicate that here to bypass the “Comments to the Author” section, enter your conflict of interest statement in the “Confidential to Editor” section, and submit your "Accept" recommendation.

Reviewer #1: All comments have been addressed

Reviewer #2: 

2. Is the manuscript technically sound, and do the data support the conclusions?

Reviewer #1: (No Response)

Reviewer #2: Yes

3. Has the statistical analysis been performed appropriately and rigorously? 

Reviewer #1: (No Response)

Reviewer #2: Yes

4. Have the authors made all data underlying the findings in their manuscript fully available?

Reviewer #1: (No Response)

Reviewer #2: Yes

5. Is the manuscript presented in an intelligible fashion and written in standard English?

Reviewer #1: (No Response)

Reviewer #2: Yes

6. Review Comments to the Author

Reviewer #1: (No Response)

Reviewer #2: This is a very interesting study and has a great deal of potential. The neuroimaging and sample size is impressive. The exposure measure however is not impressive. Even a cross section measure of blood or urine lead on a subset would have been helpful to reassure the reader that it is lead and not other correlates of poor housing (air pollution, indoor air, mold, toxic stress, racism, etc) that are driving the results. The primary concern with the manuscript is therefore that lead exposure is modeled rather than measured. Given that the confidence with which it is written with lead as the cause is very over stated. It needs to be tempered, particulary since the need for an MR scanner means that ABCD children are over represented as living in urban areas with more advanced health care systems and these urban areas may be by definition higher in lead risk. Lead may certainly be playing a role, but other factors likely are as well (perhaps synergistically with lead since the model likely captures multiple toxic exposures associated with poor, segregated housing).

Has the model been tested in a subset of ABCD children? A percentage of them must have been screened for blood lead by their physician. Is there a correlation in that subset? Was that sort of validation attempted?

While it is possible that lead risk is the cause of the results, in the absence of a measure of lead exposure and without some form of validating the model in ABCD population, this is very speculative. The way it is written ascribes too much of the findings to Lead and not to the correlates of poor, older housing which likely can’t be disentangled easily, especially without a measurement of blood lead. The wording should be more tempered.

Many of the studies listed as representing low level lead poisoning took place in the 1980’s and 1990s when low level was defined as <10 ug/dL which is not the case for today. By today’s standards these studies would represent highly exposed populations. Please revise this text to note that the definition of low level lead poisoning was higher then.

Is there an address history that allows for adjusting for moves? If not add that to limitations.

Was the NIH toolbox administered concurrent to the MRI? If so they are cross sectional. That would violate the assumptions of causal mediation analysis.

This sentence has to be amended “In the current report, the primary residential addresses of 9- to 10-year-old children were 479 used to derive community-based risk estimates of lead exposure, which our past research has 480 shown are valid proxies of exposure [28].

Stating that the authors published a study previously in ABCD using this model is not the same as validation of a model. (lines 478-480). If the lead exposure data were validated in that paper, please state how that was done, why you aren’t showing it in this paper and what the correlation of the model was with blood lead? Otherwise remove this sentence or revise it taking "valid proxies" out.

7. PLOS authors have the option to publish the peer review history of their article (what does this mean?). If published, this will include your full peer review and any attached files.

Reviewer #1: No

Reviewer #2: No

---

## [Author Response · Author response to Decision Letter 1]

14 Apr 2021

Editor

1. There continue to be important questions raised about the resubmission, as indicated by the questions and comments by the additional reviewer, particularly regarding whether there is a direct measure of blood or urine lead, even for a subset of the ABCD sample, available. This would strengthen the conclusion that it is lead and not other aspects of the poor housing that are actually responsible for the outcomes reported here. The relations between ADI and both lead risk and housing age that I inquired about were apparently not so strong as to suggest that the ADI and lead risk are one and the same and suggest that they were making relatively independent contributions. However, these analyses were based on lead “risk” and not a direct measure of lead exposure. As noted by the second reviewer, concluding that lead is the cause of the effects is overstated. Please respond to all of the points raised by this reviewer and modify the manuscript accordingly.

a. We agree that the neighborhood-level lead risk measure is only a surrogate marker for lead exposure. We reviewed the manuscript and replaced “effect” with “associations”. We also verified that we refer to “risk” of exposure rather than actual exposure throughout the manuscript. 

Reviewer #2

1. The primary concern with the manuscript is therefore that lead exposure is modeled rather than measured. Given that the confidence with which it is written with lead as the cause is very overstated. It needs to be tempered, particularly since the need for an MR scanner means that ABCD children are overrepresented as living in urban areas with more advanced health care systems and these urban areas may be, by definition, higher in lead risk. Lead may certainly be playing a role, but other factors likely are as well (perhaps synergistically with lead since the model likely captures multiple toxic exposures associated with poor, segregated housing).

b. We appreciate the reviewer’s comment and agree that we are using risk scores in contrast to blood lead concentrations. Indeed, the Methods section includes the sentence, “Currently, there are no blood-lead data for ABCD participants” (p. 6, Lines 108-109). Still, we believe our analyses are informative and provide insight into the relationships between neurocognitive development and factors associated with neurotoxicant exposure. Accordingly, our Discussion also includes the following sentence: “There are no endogenous lead-exposure data yet in ABCD, but our study suggests that lead-exposure risk is associated with corpus callosal morphology” (p. 22, Lines 421-423). 

We also added the following sentence to the Methods section: “Lead-risk scores were previously shown to be highly associated with childhood lead exposure in children (Marshall et al., 2020; D. C. Wheeler, Jones, Schootman, & Nelson, 2019). For instance, we previously validated that these lead-risk scores were positively associated with both the rate of elevated blood-lead levels across 13 states and 2 cities at the census-tract level and the geometric mean of census-tract-level blood-lead levels in the state of Maryland (Marshall et al., 2020)” (p. 6, Lines 119-123). 

Further, as noted by the reviewer, it is unlikely that the ABCD participants were only exposed to lead. Accordingly, our Discussion now includes the following sentences: “As individuals are rarely exposed to isolated chemicals (but to mixtures of chemicals) (Cory-Slechta, 2005), the incorporation of multiple data sources reflecting ‘mixtures’ of environmental health disparities may offer substantial insight into the collective and synergistic factors to target in environmental remediation interventions. Accordingly, upon collection of bodily lead data in ABCD, our future research will involve analyses of both chemical and ‘environmental-disparity mixtures’ to both study developmental trajectories more comprehensively and evaluate the relative strengths of the associations between adolescent development and other sources of disparity (e.g., air pollution, residential segregation)” (p. 27, Lines 541-548). 

As noted by the reviewer, the ABCD data collection sites were predominantly in urban areas. Still, the recruitment areas for these sites do well to encompass a non-negligible proportion of 9-to-10-year-olds in the US, which we have noted in an added sentence to the Discussion: “Similarly, even though ABCD’s data collection sites are primarily in metropolitan areas, the recruitment areas of these 21 sites represent at least 20% of the 9-to-10-year-old US population (Garavan et al., 2018)” (p. 26, Lines 510-512). 

As noted in E.1, we thoroughly reviewed the manuscript and have altered the wording of any claims that imply causation, specifically using words such as “associations” rather than “effects, and we have also reviewed the manuscript and have edited any wording to clarify that these analyses involve “risk” of exposure rather than actual exposure as measured by blood or urine collection (or other biospecimens).

2. Has the model been tested in a subset of ABCD children? A percentage of them must have been screened for blood lead by their physician. Is there a correlation in that subset? Was that sort of validation attempted?

a. We thank the reviewer for this comment, as it pertains to one of our future goals of our research. Indeed, one passage of our discussion is, “There are no endogenous lead-exposure data yet in ABCD, but our study suggests that lead-exposure risk is associated with corpus callosal morphology” (p. 22, Lines 421-423). 

We now have a sentence in our Discussion that reads, “The ABCD Study does not yet have endogenous lead-exposure data in its participants, but it has been actively collecting address histories for its participants since birth, which, when completed, will facilitate understanding of the critical developmental periods of lead neurotoxicity vulnerability” (pp. 25-26, Lines 502-508). 

We have also added a sentence to the Conclusion section: “ABCD is also exploring the collection and analysis of blood lead levels to gauge how well they are correlated with neighborhood-level lead-risk estimates” (p. 27, Lines 556-558). 

Thus, we have noted that further research in this domain will allow us to validate these lead-risk scores against current blood lead levels in ABCD children and past lead levels from deciduous tooth analysis. The reviewer also commented on past lead screening by physicians. The ABCD Study does not yet have such data, but we have added the following sentence to our Conclusion: “ABCD is also working to incorporate electronic health records as part of its dataset, which may also help elucidate these relationships via past lead screening results” (pp. 27-28, Lines 558-559).

3. While it is possible that lead risk is the cause of the results, in the absence of a measure of lead exposure and without some form of validating the model in ABCD population, this is very speculative. The way it is written ascribes too much of the findings to Lead and not to the correlates of poor, older housing which likely can’t be disentangled easily, especially without a measurement of blood lead. The wording should be more tempered.

a. Please see responses to E.1 and R2.1. We agree that neighborhood level of risk is an imperfect measure of lead exposure. Still, numerous studies have consistently shown that age of housing is a significant predictor of lead exposure using blood lead concentrations (e.g., Akkus & Ozdenerol, 2014; Benson et al., 2017; Berg, Kuhn, & Van Dyke, 2017; Binns et al., 1994; Gibson, Fisher, Clonch, MacDonald, & Cook, 2020; Haan, Gerson, & Zishka, 1996; Haley & Talbot, 2004; Kaufmann, Clouse, Olson, & Matte, 2000; Lanphear, Byrd, Auinger, & Schaffer, 1998; Litaker, Kippes, Gallagher, & O'Connor, 2000; Miranda, Dolinoy, & Overstreet, 2002; Pirkle et al., 1998; Sargent, Bailey, Simon, Blake, & Dalton, 1997; Sargent et al., 1995; Schultz, Morara, Buxton, & Weintraub, 2017; W. Wheeler & Brown, 2013).

4. Many of the studies listed as representing low level lead poisoning took place in the 1980’s and 1990s when low level was defined as <10 ug/dL which is not the case for today. By today’s standards, these studies would represent highly exposed populations. Please revise this text to note that the definition of low-level lead poisoning was higher then.

a. We appreciate the reviewer’s concern regarding how it is important to properly contextualize our data. With regard to the reviewer’s comment regarding low versus high levels of lead poisoning, our specific discussion of the relationships between “low levels” of exposure and cognition stem from 2 relatively recent papers, one from 2012 and the other from 2018, respectively: “… even at low levels of exposure, IQ was shown to be associated with concurrent blood-lead levels in 7- to 14-year-olds (Lucchini et al., 2012; Menezes-Filho et al., 2018)” (p. 25, Lines 494-495). 

To address the considerable declines in blood lead levels over the past several decades and that the criterion for elevated exposure has similarly decreased, while also reflecting evidence that blood lead levels are still relatively high in some children in the United States, we have added a sentence to the Discussion: “Indeed, while average blood lead levels have substantially declined over the past several decades, a 2021 study has estimated that nearly 400,000 1-to-11-year-olds in 2011-2016 had blood-lead levels exceeding the CDC’s reference level of 5 µg/dL (Egan, Cornwell, Courtney, & Ettinger, 2021)” (p. 24, Lines 458-461).

5. Is there an address history that allows for adjusting for moves? If not, add that to limitations.

a. We thank the reviewer for this comment. The ABCD Study is working to compile address histories for its nearly 12,000 participants. Accordingly, the Limitations section includes the following sentence: “The ABCD Study does not yet have endogenous lead-exposure data in its participants, but it has been actively collecting address histories for its participants since birth, which, when completed, will facilitate understanding of the critical developmental periods of lead neurotoxicity vulnerability” (pp. 25-26, Lines 502-508). 

Further, the Conclusion now includes the following sentences: “ABCD is also exploring the collection and analysis of blood lead levels to gauge how well they are correlated with neighborhood-level lead-risk estimates. ABCD is also working to incorporate electronic health records as part of its dataset, which may also help elucidate these relationships via past lead screening results” (pp. 27-28, Lines 554-557). Specifically, while the ABCD Study is working to include full address histories for its participants, this has not yet been completed given the rigor and effort to complete this task. Even though research has shown that characteristics individuals’ neighborhoods are often relatively stable over time, even when participants move (Knighton, 2018), and even across generations (van Ham, Hedman, Manley, Coulter, & Östh, 2014), we have now noted in the manuscript our intention to analyze such address histories once these data are made available.

6. Was the NIH toolbox administered concurrent to the MRI? If so, they are cross sectional. That would violate the assumptions of causal mediation analysis.

a. The reviewer noted that mediational analyses of cross-sectional data are not viable given such cross-sectional design. As described by Fairchild and McDaniel (2017), this guideline is generally true, but it is possible to run mediational analyses on cross-sectional data given the analytical design. Specifically, cross-sectional mediational analyses are possible if the correct temporal order is established and that alternative temporal orderings are ruled out (Fairchild & McDaniel, 2017). We have edited our Limitations section to now include the following sentence: “Further, while the potential for reverse causation is inherent to cross-sectional studies, thereby limiting causal inference, it is unlikely that poor cognitive performance elicits altered brain structure, or that altered brain structure induces risk factors of lead exposure here, thus supporting the temporal ordering within our cross-sectional mediational analyses (lead risk � brain structure � cognitive performance) (Fairchild & McDaniel, 2017)” (p. 26, Lines 510-514).

7. This sentence has to be amended “In the current report, the primary residential addresses of 9- to 10-year-old children were used to derive community-based risk estimates of lead exposure, which our past research has shown are valid proxies of exposure [28]. Stating that the authors published a study previously in ABCD using this model is not the same as validation of a model. (lines 478-480). If the lead exposure data were validated in that paper, please state how that was done, why you aren’t showing it in this paper and what the correlation of the model was with blood lead? Otherwise remove this sentence or revise it taking "valid proxies" out.

As also noted in R2.1, we have also added the following sentence to the Methods section to further describe the lead-risk model developed by Rad Cunningham at the Washington State Department of Health: “Lead-risk scores were previously shown to be highly associated with childhood lead exposure in children (Marshall et al., 2020; D. C. Wheeler et al., 2019). For instance, we previously validated that these lead-risk scores were positively associated with both the rate of elevated blood-lead levels across 13 states and 2 cities at the census-tract level and the geometric mean of census-tract-level blood-lead levels in the state of Maryland (Marshall et al., 2020)” (p. 6, Lines 119-123). Because these data have been published elsewhere, we chose to provide more detail to describe what we had done previously without replicating the previous report.

References

Akkus, C., & Ozdenerol, E. (2014). Exploring childhood lead exposure through GIS: A review of the recent literature. International Journal of Environmental Research and Public Health, 11, 6314-6334. doi:10.3390/ijerph110606314

Benson, S. M., Talbott, E. O., Brink, L. L., Wu, C., Sharma, R. K., & Marsh, G. M. (2017). Environmental lead and childhood blood lead levels in US children: NHANES, 1999-2006. Archives of Environmental & Occupational Health, 72(2), 70-78. doi:10.1080/19338244.2016.1157454

Berg, K., Kuhn, S., & Van Dyke, M. (2017). Spatial surveillance of childhood lead exposure in a targeted screening state: An application of generalized additive models in Denver, Colorado. Journal of Public Health Management & Practice, 23, S79-S92. 

Binns, H. J., LeBailly, S. A., Poncher, J., Kinsella, R., Saunders, S. E., & Pediatric Practice Research Group. (1994). Is there lead in the suburbs? Risk assessment in Chicago suburban pediatric practices. Pediatrics, 93(2), 164-171. 

Cory-Slechta, D. A. (2005). Studying toxicants as single chemicals: Does this strategy adequately identify neurotoxic risk? NeuroToxicology, 26(4), 491-510. doi:10.1016/j.neuro.2004.12.007

Egan, K. B., Cornwell, C. R., Courtney, J. G., & Ettinger, A. S. (2021). Blood lead levels in U.S. children ages 1-11 years, 1976-2016. Environmental Health Perspectives, 129(3), 037003. doi:10.1289/EHP7932

Fairchild, A. J., & McDaniel, H. L. (2017). Best (but oft-forgotten) practices: mediation analysis. American Journal of Clinical Nutrition, 105, 1259-1271. doi:10.3945/ajcn.117.152546

Garavan, H., Bartsch, H., Conway, K., Decastro, A., Goldstein, R. Z., Heeringa, S., . . . Zahs, D. (2018). Recruiting the ABCD sample: Design considerations and procedures. Developmental Cognitive Neuroscience, 32, 16-22. doi:10.1016/j.dcn.2018.04.004

Gibson, J. M., Fisher, M., Clonch, A., MacDonald, J. M., & Cook, P. J. (2020). Children drinking private well water have higher blood lead than those with city water. Proceedings of the National Academy of Sciences, 117(29), 16898-16907. doi:10.1073/pnas.2002729117

Haan, M. N., Gerson, M., & Zishka, B. A. (1996). Identification of children at risk of lead poisoning: An evaluation of routine pediatric blood lead screening in an HMO-insured population. Pediatrics, 97(1), 79-83. 

Haley, V. B., & Talbot, T. O. (2004). Geographic analysis of blood lead levels in New York state children born 1994-1997. Environmental Health Perspectives, 112, 1577-1582. doi:10.1289/ehp.7053

Kaufmann, R. B., Clouse, T. L., Olson, D. R., & Matte, T. D. (2000). Elevated blood lead levels and blood lead screening among US children aged one to five years: 1988-1994. Pediatrics, 106, e79. 

Knighton, A. J. (2018). Is a patient's current address of record a reasonable measure of neighborhood deprivation exposure? A case for the use of point in time measures of residence in clinical care. Health Equity, 2, 62-69. doi:10.1089/heq.2017.0005

Lanphear, B. P., Byrd, R. S., Auinger, P., & Schaffer, S. J. (1998). Community characteristics associated with elevated blood lead levels in children. Pediatrics, 101(2), 264-271. 

Litaker, D., Kippes, C. M., Gallagher, T. E., & O'Connor, M. E. (2000). Targeting lead screening: The Ohio lead risk score. Pediatrics, 106(5), e69. 

Lucchini, R. G., Zoni, S., Guazzetti, S., Bontempi, E., Micheletti, S., Broberg, K., . . . Smith, D. R. (2012). Inverse association of intellectual function with very low blood lead but not with manganese exposure in Italian adolescents. Environmental Research, 118, 65-71. doi:10.1016/j.envres.2012.08.003

Marshall, A. T., Betts, S., Kan, E. C., McConnell, R., Lanphear, B. P., & Sowell, E. R. (2020). Association of lead-exposure risk and family income with childhood brain outcomes. Nature Medicine, 26, 91-97. doi:10.1038/s41591-019-0713-y

Menezes-Filho, J. A., Carvalho, C. F., Rodrigues, J. L. G., Araújo, C. F. S., dos Santos, N. R., Lima, C. S., . . . Mergler, D. (2018). Environmental co-exposure to lead and manganese and intellectual deficit in school-aged children. International Journal of Environmental Research and Public Health, 15, 2418. doi:10.3390/ijerph15112418

Miranda, M. L., Dolinoy, D. C., & Overstreet, M. A. (2002). Mapping for prevention: GIS models for directing childhood lead poisoning prevention programs. Environmental Health Perspectives, 110, 947-953. 

Pirkle, J. L., Kaufmann, R. B., Brody, D. J., Hickman, T., Gunter, E. W., & Paschal, D. C. (1998). Exposure of the U.S. population to lead, 1991-1994. Environmental Health Perspectives, 106(11), 745-790. doi:10.1289/ehp.98106745

Sargent, J. D., Bailey, A., Simon, P., Blake, M., & Dalton, M. A. (1997). Census tract analysis of lead exposure in Rhode Island children. Environmental Research, 74, 159-168. 

Sargent, J. D., Brown, M. J., Freeman, J. L., Bailey, A., Goodman, D., & Freeman, D. H. (1995). Childhood lead poisoning in Massachusetts communities: Its association with sociodemographic and housing characteristics. American Journal of Public Health, 85, 528-534. 

Schultz, B. D., Morara, M., Buxton, B. E., & Weintraub, M. (2017). Predicting blood-lead levels amoung U.S. children at the census tract level. Environmental Justice, 10, 129-136. doi:10.1089/env.2017.0005

van Ham, M., Hedman, L., Manley, D., Coulter, R., & Östh, J. (2014). Intergenerational transmission of neighbourhood poverty: an analysis of neighbourhood histories of individuals. Transactions of the Institute of British Geographers, 39, 402-417. doi:10.1111/tran.12040

Wheeler, D. C., Jones, R. M., Schootman, M., & Nelson, E. J. (2019). Explaining variation in elevated blood lead levels among children in Minnesota using neighborhood socioeconomic variables. Science of the Total Environment, 650, 970-977. doi:10.1016/j.scitotenv.2018.09.088

Wheeler, W., & Brown, M. J. (2013). Blood lead levels in children aged 1-5 years - United States, 1999-2010. Morbidity and Mortality Weekly Report, 62(13), 245-248.

---

## [Decision Letter · Decision Letter 2]

16 Sep 2021

PONE-D-20-37575R2

Risk of lead exposure, subcortical brain structure, and cognition in a large cohort of 9- to 10-year-old children

PLOS ONE

Dear Dr. Marshall,

Thank you for submitting your revised manuscript to PLOS ONE. After careful consideration, we feel that it has merit but does not fully meet PLOS ONE’s publication criteria as it currently stands. Therefore, we invite you to submit a revised version of the manuscript that addresses the points raised during the review process.

The interest in and significance of research on effects of lead exposure is major and has many implications for public health. This paper focuses on lead risk but lacks direct biological measures of lead exposure. This important limitation results in the authors’ needing to expand on this limitation rather than concluding with lead exposure (not lead risk) being the primary cause of the developmental problems.  Moreover, the sentence in the Conclusion is incorrect: "Lead exposure may mediate certain aspects of cognitive functioning by diminishing subcortical brain structure, including the anterior splenium (i.e., mid-posterior corpus callosum)." Lead is not the mediator in this sentence but the predictor; regional brain volume is the mediator." This is correctly stated in a sentence towards the end of the previous paragraph.

One reviewer has concluded that the paper is acceptable now. However, the other reviewer points out some important limitations that still need to be addressed. Even though the authors adjusted for environmental factors, statistical control may still be inadequate, particularly given the absence of a biological measure of lead. Thus, as pointed out by the second reviewer, in their discussion, the authors need to embed lead risk into the larger environmental risk factors that contribute to the outcome, of which lead is only one. The authors should, therefore, modify the discussion and conclusions to emphasize the combination of factors that impact on the public health of the children.

We look forward to receiving your revised manuscript.

Kind regards,

Sandra Jacobson

Academic Editor

PLOS ONE

Journal Requirements:

Additional Editor Comments (if provided):

Reviewers' comments:

Reviewer's Responses to Questions

**Comments to the Author**

1. If the authors have adequately addressed your comments raised in a previous round of review and you feel that this manuscript is now acceptable for publication, you may indicate that here to bypass the “Comments to the Author” section, enter your conflict of interest statement in the “Confidential to Editor” section, and submit your "Accept" recommendation.

Reviewer #1: All comments have been addressed

Reviewer #2: (No Response)

2. Is the manuscript technically sound, and do the data support the conclusions?

Reviewer #1: (No Response)

Reviewer #2: Yes

3. Has the statistical analysis been performed appropriately and rigorously? 

Reviewer #1: (No Response)

Reviewer #2: Yes

4. Have the authors made all data underlying the findings in their manuscript fully available?

Reviewer #1: (No Response)

Reviewer #2: (No Response)

5. Is the manuscript presented in an intelligible fashion and written in standard English?

Reviewer #1: (No Response)

Reviewer #2: Yes

6. Review Comments to the Author

Reviewer #1: (No Response)

Reviewer #2: “Lead-risk scores were previously shown to be highly associated with childhood lead exposure in children (Marshall et al., 2020; D. C. Wheeler, Jones, Schootman, & Nelson, 2019). For instance, we previously validated that these lead-risk scores were positively associated with both the rate of elevated blood-lead levels across 13 states and 2 cities at the census-tract level and the geometric mean of census-tract-level blood-lead levels in the state of Maryland (Marshall et al., 2020)” (p. 6, Lines 119-123)

What is described above is an ecological analysis and not a validation study (i.e. the children in the analysis cited for validation may or may not be the children screened for lead). More to the point- the authors need consideration of the issues that correlate with lead exposure that may be playing a bigger role than lead. The reader can decide if they believe the results are due to lead or other correlates of poor housing. You haven’t proven it is lead, and with the limitations you have on measuring lead exposure, can never prove it is lead. Just be transparent about that. It may be lead in part, I don’t dispute that, but there is a lack of consideration of all the issues that go with poor housing- especially racism, economic stress and educational quality, just to name a few.

This paper needs greater discussion on the role that segregated housing and redlining is playing as I believe those issues are severely understated and the narrow focus on lead exposure does a disservice to children. Focusing on lead poisoning also misses the big picture which is structural racism. The bulk of these findings are due to the combination of issues(redlining, economic injustice and social stress) and not just lead. In addition, lead poisoning didn’t cause redlining and segregation, it is a consequence of it. Fixing lead exposure won’t fix segregation and its many toxic correlates. Fixing segregation will fix the larger issues, including lead exposure and will have much greater public health impact than just focusing on lead exposure. The discussion should address these issues as give them greater weight.

7. PLOS authors have the option to publish the peer review history of their article (what does this mean?). If published, this will include your full peer review and any attached files.

Reviewer #1: No

Reviewer #2: No

---

## [Author Response · Author response to Decision Letter 2]

21 Sep 2021

Editor

1. The interest in and significance of research on effects of lead exposure is major and has many implications for public health. This paper focuses on lead risk but lacks direct biological measures of lead exposure. This important limitation results in the authors’ needing to expand on this limitation rather than concluding with lead exposure (not lead risk) being the primary cause of the developmental problems. Moreover, the sentence in the Conclusion is incorrect: "Lead exposure may mediate certain aspects of cognitive functioning by diminishing subcortical brain structure, including the anterior splenium (i.e., mid-posterior corpus callosum)." Lead is not the mediator in this sentence but the predictor; regional brain volume is the mediator." This is correctly stated in a sentence towards the end of the previous paragraph.

a. We thank the editor for making this comment, as that sentence in the Conclusion of the Abstract was indeed misworded. Accordingly, the Conclusion of the Abstract was edited and now reads, “Environmental factors related to the risk of lead exposure may be associated with certain aspects of cognitive functioning via diminished subcortical brain structure, including the anterior splenium (i.e., mid-posterior corpus callosum).” Additionally, in the Conclusion section at the end of the Discussion, the final sentence now reads, “Until then, this study, which uses neighborhood-level lead-exposure risk, provides potential evidence that cognitive deficits from low-level lead toxicity (and its related environmental factors) may operate by diminishing subcortical brain structure [28]” (pp. 28-29, Lines 567-570).

2. One reviewer has concluded that the paper is acceptable now. However, the other reviewer points out some important limitations that still need to be addressed. Even though the authors adjusted for environmental factors, statistical control may still be inadequate, particularly given the absence of a biological measure of lead. Thus, as pointed out by the second reviewer, in their discussion, the authors need to embed lead risk into the larger environmental risk factors that contribute to the outcome, of which lead is only one. The authors should, therefore, modify the discussion and conclusions to emphasize the combination of factors that impact on the public health of the children.

a. As also described in R2.1, the primary revisions regarded contextualizing risk of lead exposure within other environmental variables. We thank the editor and Reviewer #2 for addressing this comment, as we agree that risk of lead exposure does not operate in a vacuum but is associated with other socioeconomic and demographic factors. Accordingly, we have edited the Limitations section to include the following paragraph, which we believe effectively addresses the greater context of lead exposure: “As individuals are rarely exposed to isolated chemicals (but to mixtures of chemicals) [104], the incorporation of multiple data sources reflecting “mixtures” of environmental health disparities may also offer substantial insight into the collective and synergistic factors to target in environmental remediation interventions. Simply, the risk of lead exposure does not exist in a vacuum but is associated with past and current practices that have differentially subjected children to such risks. For example, the burden of lead-exposure’s effects is typically greatest in children in the lowest SES families [105-108], a glaring example of environmental injustice [105, 109]. Similarly, Black and Hispanic children tend to have greater mean blood lead levels than white children [5, 110-112] and are more likely than white children to live in homes or regions with greater risks of lead exposure [113-115]. Further, lead-poisoning rates (and, thus, children’s blood-lead levels) are associated with multiple community-level factors [39], including value and age of houses, poverty rates, population density, and percentage of the population who are Black or Hispanic [1, 116], signifying racial residential segregation as a potential explanatory mechanism for lead exposure disparities [117]. While the data in the current manuscript may reflect differences in lead exposure, these differences would then ultimately be due to disparate conditions that initially elicited such differences, thereby focusing any potential intervention efforts on the originating disparities. Indeed, recent research has shown that soil-lead concentrations tended to be elevated in samples taken from historically redlined neighborhoods compared to those in “best” and “desirable” neighborhoods, per zone designations by the 1930’s Home Owners’ Loan Corporation [118]. Ultimately, because we do not currently have endogenous lead-exposure data in our participants, the results related to risk of lead exposure may be alternatively explained (at least partially) by these other systemic and environmental factors. Accordingly, upon collection of bodily lead data and additional geocoded data in ABCD, our future research will involve analyses of both chemical and “environmental-disparity mixtures” to both study developmental trajectories more comprehensively and evaluate the relative strengths of the associations between adolescent development and other sources of disparity (e.g., air pollution, residential segregation)” (pp. 27-28, Lines 530-556). Further, throughout the discussion, we have also edited sentences to not only describe lead-exposure risk, but also its related environmental factors that may also account for our results (p. 22, Lines 427-428; p. 24, Line 468).

Reviewer #2

1. Lead-risk scores were previously shown to be highly associated with childhood lead exposure in children (Marshall et al., 2020; D. C. Wheeler, Jones, Schootman, & Nelson, 2019). For instance, we previously validated that these lead-risk scores were positively associated with both the rate of elevated blood-lead levels across 13 states and 2 cities at the census-tract level and the geometric mean of census-tract-level blood-lead levels in the state of Maryland (Marshall et al., 2020)” (p. 6, Lines 119-123) What is described above is an ecological analysis and not a validation study (i.e. the children in the analysis cited for validation may or may not be the children screened for lead). More to the point- the authors need consideration of the issues that correlate with lead exposure that may be playing a bigger role than lead. The reader can decide if they believe the results are due to lead or other correlates of poor housing. You haven’t proven it is lead, and with the limitations you have on measuring lead exposure, can never prove it is lead. Just be transparent about that. It may be lead in part, I don’t dispute that, but there is a lack of consideration of all the issues that go with poor housing- especially racism, economic stress and educational quality, just to name a few. This paper needs greater discussion on the role that segregated housing and redlining is playing as I believe those issues are severely understated and the narrow focus on lead exposure does a disservice to children. Focusing on lead poisoning also misses the big picture which is structural racism. The bulk of these findings are due to the combination of issues (redlining, economic injustice and social stress) and not just lead. In addition, lead poisoning didn’t cause redlining and segregation, it is a consequence of it. Fixing lead exposure won’t fix segregation and its many toxic correlates. Fixing segregation will fix the larger issues, including lead exposure and will have much greater public health impact than just focusing on lead exposure. The discussion should address these issues as give them greater weight.

a. We appreciate the reviewer’s comment on validation studies versus ecological studies. Accordingly, when we first introduce the lead-risk and area-deprivation indices in the Methods section, we have altered the wording of the sentence describing the validation of the lead-risk scores. Specifically, the sentence now reads, “For instance, we previously showed that these lead-risk scores were positively associated with both the rate of elevated blood-lead levels across 13 states and 2 cities at the census-tract level and the geometric mean of census-tract-level blood-lead levels in the state of Maryland [28]” (p. 6, Line 125). Further, as described in the response to E.2, we have edited the Limitations section to include the following paragraph, which we believe effectively addresses the greater context of lead exposure: “As individuals are rarely exposed to isolated chemicals (but to mixtures of chemicals) [104], the incorporation of multiple data sources reflecting “mixtures” of environmental health disparities may also offer substantial insight into the collective and synergistic factors to target in environmental remediation interventions. Simply, the risk of lead exposure does not exist in a vacuum but is associated with past and current practices that have differentially subjected children to such risks. For example, the burden of lead-exposure’s effects is typically greatest in children in the lowest SES families [105-108], a glaring example of environmental injustice [105, 109]. Similarly, Black and Hispanic children tend to have greater mean blood lead levels than white children [5, 110-112] and are more likely than white children to live in homes or regions with greater risks of lead exposure [113-115]. Further, lead-poisoning rates (and, thus, children’s blood-lead levels) are associated with multiple community-level factors [39], including value and age of houses, poverty rates, population density, and percentage of the population who are Black or Hispanic [1, 116], signifying racial residential segregation as a potential explanatory mechanism for lead exposure disparities [117]. While the data in the current manuscript may reflect differences in lead exposure, these differences would then ultimately be due to disparate conditions that initially elicited such differences, thereby focusing any potential intervention efforts on the originating disparities. Indeed, recent research has shown that soil-lead concentrations tended to be elevated in samples taken from historically redlined neighborhoods compared to those in “best” and “desirable” neighborhoods, per zone designations by the 1930’s Home Owners’ Loan Corporation [118]. Ultimately, because we do not currently have endogenous lead-exposure data in our participants, the results related to risk of lead exposure may be alternatively explained (at least partially) by these other systemic and environmental factors. Accordingly, upon collection of bodily lead data and additional geocoded data in ABCD, our future research will involve analyses of both chemical and “environmental-disparity mixtures” to both study developmental trajectories more comprehensively and evaluate the relative strengths of the associations between adolescent development and other sources of disparity (e.g., air pollution, residential segregation)” (pp. 27-28, Lines 530-556). Further, throughout the discussion, we have also edited sentences to not only describe lead-exposure risk, but also its related environmental factors that may also account for our results (p. 22, Lines 427-428; p. 24, Line 468).

---

## [Editor Report · Decision Letter 3]

29 Sep 2021

Risk of lead exposure, subcortical brain structure, and cognition in a large cohort of 9- to 10-year-old children

PONE-D-20-37575R3

Dear Dr. Marshall,

We are pleased to inform you that your manuscript has been carefully reviewed and is now considered suitable for publication and will be formally accepted for publication once it meets all outstanding technical requirements.

Within 1 week, you will receive an e-mail detailing the required amendments. When these have been addressed, you will receive a formal acceptance letter and your manuscript will be scheduled for publication.

Kind regards,

Sandra W Jacobson

Academic Editor

PLOS ONE
---

## [Editor Report · Acceptance letter]

5 Oct 2021

PONE-D-20-37575R3 

Risk of lead exposure, subcortical brain structure, and cognition in a large cohort of 9- to 10-year-old children 

Dear Dr. Marshall:

I'm pleased to inform you that your manuscript has been deemed suitable for publication in PLOS ONE. Congratulations! Your manuscript is now with our production department. 

Kind regards, 

on behalf of

Dr. Sandra Jacobson 

Academic Editor

PLOS ONE